# BRIDGING PIANO TRANSCRIPTION AND RENDERING VIA DISENTANGLED SCORE CONTENT AND STYLE

**Wei Zeng[1,2], Junchuan Zhao[2], Ye Wang[1,2]**
[1]Integrative Sciences and Engineering Programme, NUS Graduate School
[2]School of Computing
National University of Singapore
{w.zeng,junchuan}@u.nus.edu, dcswangy@nus.edu.sg

## ABSTRACT

Expressive performance rendering (EPR) and automatic piano transcription (APT) are fundamental yet inverse tasks in music information retrieval: EPR generates expressive performances from symbolic scores, while APT recovers scores from performances. Despite their dual nature, prior work has addressed them independently. In this paper, we propose a unified framework that jointly models EPR and APT by disentangling note-level score content and global performance style representations from both paired and unpaired data. Our framework is built on a transformer-based sequence-to-sequence (Seq2Seq) architecture and is trained using only sequence-aligned data, without requiring fine-grained note-level alignment. To automate the rendering process while ensuring stylistic compatibility with the score, we introduce an independent diffusion-based performance style recommendation (PSR) module that generates style embeddings directly from score content. This modular component supports both style transfer and flexible rendering across a range of expressive styles. Experimental results from both objective and subjective evaluations demonstrate that our framework achieves competitive performance on EPR and APT tasks, while enabling effective content–style disentanglement, reliable style transfer, and stylistically appropriate rendering. Demos are available at `https://wei-zeng98.github.io/joint-apt-epr/`.

## 1 INTRODUCTION

Music exists across multiple modalities, notably symbolic music scores and expressive audio recordings. Converting between these musical modalities is essential for artistic performance and musical analysis, supporting a wide range of applications from artistic creation to music education (Cancino-Chacón et al., 2023; Chacón et al., 2023). In a live concert, for example, a pianist *renders* a written score into an expressive performance, adding personalized nuances in timing, dynamics, and articulation. Conversely, for purposes such as analysis, re-performance, or archiving, *transcription* is needed to convert an audio recording of a performance back into a symbolic representation. These two processes correspond to two core tasks in music information retrieval (MIR): expressive performance rendering (EPR), which generates performance MIDI (MIDI that captures expressive timing, dynamics, and articulation) from symbolic scores (Chacón et al., 2018), and automatic piano transcription (APT), which predicts symbolic scores from performance MIDI (Desain & Honing, 1989).

Prior work has studied EPR and APT as two separate tasks (Maezawa et al., 2019; Jeong et al., 2019; Rhyu et al., 2022; Borovik & Viro, 2023; Liu et al., 2022; Cogliati et al., 2016; Nakamura et al., 2018; Shibata et al., 2021). However, as illustrated in the top-left corner of Figure 1, the two tasks are inherently connected, representing inverse transformations between symbolic and expressive forms. In rendering, the performance reflects both the composer's intent and the pianist's interpretive style; in transcription, the system should filter out these expressive elements to recover the underlying score.

Joint modeling in speech tasks such as automatic speech recognition (ASR) and text-to-speech (TTS) has shown mutual benefits and enabled weakly supervised training (Ren et al., 2019; Peyser et al., 2022a). A concurrent line of work in music demonstrates a similar direction, showing that unified translation across multiple modalities can be achieved using only sequence-aligned data (Jung et al.,

2025), underscoring the growing trend toward scalable, alignment-free supervision. Motivated by these developments, we propose a unified transformer-based framework that jointly learns EPR and APT by modeling two factors: (a) a note-level score content representation, which captures symbolic structures like pitch and rhythm; and (b) a global performance style representation, which encapsulates the high-level artistic character of a performance (e.g., "heavy" or "relaxing") and serves as a conditioning signal to guide the generation of fine-grained expressive details by the decoder. This disentangled representation allows for information sharing across tasks while preserving the interpretability and controllability of the rendering process. Furthermore, the use of a unified Seq2Seq architecture enables our model to be trained using only sequence-aligned data, removing the need for note-level alignment required by most EPR systems (Rhyu et al., 2022; Borovik & Viro, 2023; Tang et al., 2023; Jeong et al., 2019; Zhang et al., 2024).

To enable flexible and realistic performance rendering, it is crucial to distinguish between the types of information encoded in our disentangled representations. We define *style* as the expressive realization of a score (e.g., the "Horowitz factor" by Widmer et al. (2003)), and *genre* as the underlying structural and harmonic characteristics of the composition. While both are global attributes, they capture distinct musical aspects. Inspired by recent advances in sheet music classification (Ji et al., 2021; Pasquale et al., 2020), we hypothesize that for a performance rendition to sound natural, the chosen style should ideally align with the underlying genre. This suggests that stylistically appropriate performances can be inferred directly from score content, similar to how skilled pianists interpret compositions. Moreover, existing EPR models often rely on composer labels (Jeong et al., 2019; Tang et al., 2023) or require manual control over expressive parameters (Borovik & Viro, 2023; Rhyu et al., 2022), which limits accessibility for non-expert users. Motivated by these observations, we propose a Performance Style Recommendation (PSR) module that generates diverse style embeddings conditioned solely on the score.

We evaluate our framework using both objective and subjective metrics. On standard benchmarks, our joint model achieves competitive performance for both EPR and APT. Subjective evaluations confirm the naturalness of EPR-generated performances. Disentanglement is verified through style transfer and latent space visualizations. In addition, we show that the learned style embeddings encode information about both performer and composer, with composer traits being more dominant. Finally, evaluations of the PSR module demonstrate its ability to generate stylistically appropriate embeddings from content alone.

In summary, this paper makes the following three contributions:

- **A unified transformer-based model for joint EPR and APT**, which disentangles score content and performance style representations, and leverages the duality between the two tasks for mutual supervision. This joint formulation enables bidirectional modeling between symbolic and expressive forms of music.

- **A diffusion-based performance style recommendation (PSR) module**, which generates diverse and appropriate style embeddings directly from score content. This module mimics a pianist's ability to infer suitable expressive styles from a written score, enabling controllable rendering and making style-aware performance accessible to non-experts.

- **A Seq2Seq formulation of EPR without note-level alignment**, which eliminates the need for finely aligned training data and enables scalable learning using only sequence-level supervision. Despite this relaxed supervision, our model achieves competitive performance compared to alignment-dependent baselines.

## 2 RELATED WORK

### 2.1 EXPRESSIVE PIANO PERFORMANCE RENDERING

Early work on EPR relied on rule-based systems (Widmer & Goebl, 2004; Chacón et al., 2018; Kirke & Miranda, 2013). Recent methods leverage deep learning, including RNN- and LSTM-based models (Maezawa et al., 2019; Jeong et al., 2019), as well as transformer-based architectures (Rhyu et al., 2022; Borovik & Viro, 2023; Renault et al., 2023; Tang et al., 2023). A central challenge in EPR is generating performance styles that appropriately reflect the content of music scores. Existing approaches often require explicit composer or performer labels (Jeong et al., 2019; Tang et al., 2023),

or depend on manual control of expressive parameters (Borovik & Viro, 2023; Rhyu et al., 2022), limiting usability for non-expert users. A diffusion-based model has been introduced to generate expressive control directly from the score, relying on hand-crafted note-level style features (Zhang et al., 2024). However, such a note-level approach demands intricate, fine-grained adjustments and offers limited flexibility for style transfer between compositions with disparate musical structures.

Another key limitation of current models (Rhyu et al., 2022; Borovik & Viro, 2023; Tang et al., 2023; Jeong et al., 2019; Zhang et al., 2024) is their dependence on note-aligned datasets, which typically require preprocessing with alignment tools (Nakamura et al., 2017). This reliance impedes flexibility, particularly for expressive techniques like trills and mordents that introduce temporal ambiguity. An unsupervised GAN-based approach has been proposed to bypass alignment (Renault et al., 2023), but it is less performant than supervised counterparts. Recent work also explores sequence-aligned supervision as a scalable alternative; for instance, Jung et al. (2025) demonstrate that unified cross-modal music translation can be effectively learned without strict note-level alignment. Motivated by these developments, we address these limitations by formulating EPR as a Seq2Seq task and introducing a PSR module for automatic style generation.

## 2.2 AUTOMATIC PIANO TRANSCRIPTION

Automatic piano transcription (APT) methods can be categorized by their input and output modalities. Input formats include raw audio signals (e.g., waveforms or spectrograms) and symbolic representations such as MIDI. Output targets are typically note-level sequences (Hawthorne et al., 2018; Kim & Bello, 2019; Kong et al., 2021; Toyama et al., 2023; Hawthorne et al., 2021) or notation-level formats resembling human-readable sheet music (Román et al., 2019; Alfaro-Contreras et al., 2024; Román et al., 2018; Zeng et al., 2024; Hiramatsu et al., 2021; Liu et al., 2021; 2022; Shibata et al., 2021; Beyer & Dai, 2024). This work focuses on symbolic-to-symbolic transcription, where the model maps expressive performance MIDI to corresponding score sheet representations.

Early APT approaches relied on signal processing heuristics (Raphael, 2001) and probabilistic models such as Hidden Markov Models (HMMs) (Cogliati et al., 2016; Shibata et al., 2021). Recent advances leverage deep neural networks (Liu et al., 2022; Beyer & Dai, 2024; Suzuki, 2021), which have demonstrated substantial improvements in accuracy and generalization. Particularly, (Beyer & Dai, 2024) proposed a Seq2Seq framework that eliminates the need for note-aligned supervision while achieving state-of-the-art performance. Building on this insight, we adopt a similar Seq2Seq framework to model score content features within our unified system.

## 2.3 DISENTANGLED REPRESENTATION LEARNING

Disentangled representation learning (DRL) aims to learn representations that separate the underlying factors of variation in observed data (Wang et al., 2024). It has been widely studied in computer vision (Dupont, 2018; Yang et al., 2021; Chen et al., 2016; Karras et al., 2020) and natural language processing (He et al., 2017; Bao et al., 2019; Cheng et al., 2020; Wu et al., 2020), where separating content from style or semantics has led to improved generalization and controllability.

In music information retrieval (MIR), DRL has recently been explored for disentangling musical content and style to support generation and manipulation (Tan & Herremans, 2020; Wang et al., 2020; Yang et al., 2019; Zhao et al., 2024). One closely related study (Zhang & Dixon, 2023) learns content and style representations from expressive performances in an unsupervised manner, enabling music analysis and style transfer. In contrast, our work focuses on generating expressive performances from symbolic scores, a less-explored but important direction for DRL-based music modeling.

## 3 METHODOLOGY

### 3.1 DATA REPRESENTATION FOR INPUT AND OUTPUT

**Input features**   Following Peyser et al. (2022b), we represent both score and performance inputs as note-level sequences of approximately equal length, enabling the joint encoder to learn a domain-agnostic representation of score content. We denote the score and performance sequences as $\mathbf{x}$ and $\mathbf{y}$, respectively. Each sequence contains $N$ notes by the order of onset time and pitch, with each note

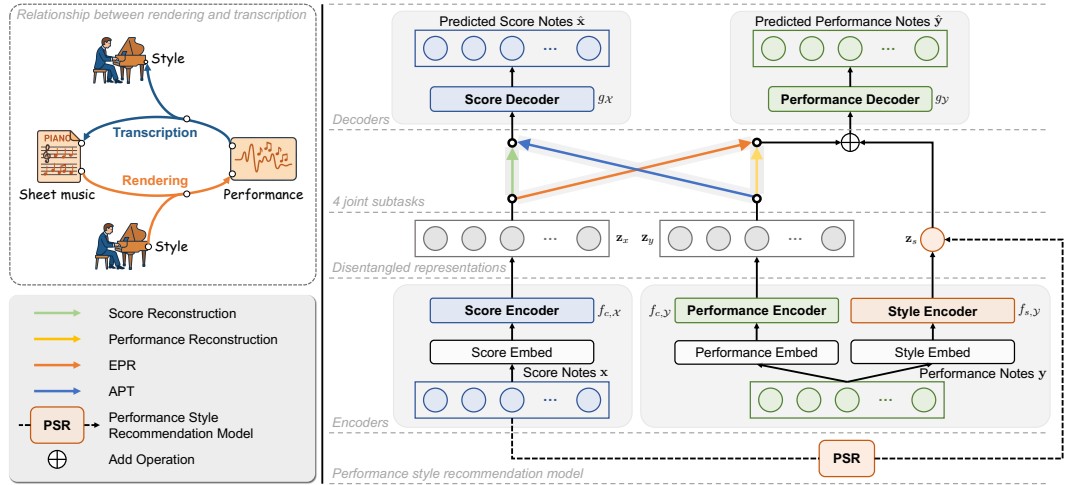

Figure 1: Relationship between EPR and APT (top left) and an overview of the proposed framework. The model comprises a joint transformer-based architecture for EPR and APT, along with a diffusion-based performance style recommendation (PSR) module. Four tasks are trained jointly: masked score reconstruction, masked performance reconstruction, expressive performance rendering (EPR), and automatic piano transcription (APT). Score content features $\mathbf{z}_x$ and $\mathbf{z}_y$, extracted from score and performance inputs respectively, are encouraged to align. A global style feature $\mathbf{z}_s$ is learned as a disentangled factor to support style transfer. The PSR module is *independently* trained to generate $\mathbf{z}_s$ from score content alone, emulating a pianist's ability to select appropriate performance styles.

represented as a tuple of $k_i$ ($i \in \{x, y\}$) discrete symbolic attributes, detailed in Appendix A.2. For score inputs, each note comprises $k_x = 8$ attributes, while performance inputs contain $k_y = 4$. The final note embedding is obtained by summing the embeddings of its constituent attributes, resulting in $\mathbf{E}_x, \mathbf{E}_y \in \mathbb{R}^{N \times D}$, where $D$ denotes the embedding dimension.

**Output features** For score prediction ($\hat{\mathbf{x}}$), we adopt the representation scheme introduced in Beyer & Dai (2024). For performance prediction ($\hat{\mathbf{y}}$), we initially applied the same tokenization as used in the input representation, but observed that it degraded generation quality. Since our Seq2Seq model does not require note-level alignment, we instead adopt the structured performance representation proposed in Huang & Yang (2020), implemented via the MidiTok library (Fradet et al., 2021).

### 3.2 UNIFIED MODELING OF EPR AND APT

We consider two domains of symbolic musical sequences: score sequences $\mathbf{x} \in \mathcal{X}$ and performance sequences $\mathbf{y} \in \mathcal{Y}$. These two domains are connected by two inverse processes: expressive performance rendering (EPR), mapping scores to performances ($\mathcal{X} \to \mathcal{Y}$), and automatic piano transcription (APT), mapping performances to scores ($\mathcal{Y} \to \mathcal{X}$). Both domains share a latent content space $\mathcal{Z}_c$, capturing note-level attributes such as pitch and rhythm. Formally, we treat EPR as a **one-to-many** mapping; in addition to content, $\mathcal{Y}$ depends on a style space $\mathcal{Z}_s$, serving as a conditioning signal for the overall control of its performance style. Conversely, APT is formulated as a **many-to-one** task, where the system aims to filter out performance-specific expressive nuances to recover the unique score content represented in $\mathcal{Z}_c$. Our framework supports training on both paired and unpaired data.

**Paired setting** Given paired data $(\mathbf{x}, \mathbf{y})$, we define content encoders $f_{c,\mathcal{X}} : \mathcal{X} \to \mathcal{Z}_c$ and $f_{c,\mathcal{Y}} : \mathcal{Y} \to \mathcal{Z}_c$, along with a style encoder $f_{s,\mathcal{Y}} : \mathcal{Y} \to \mathcal{Z}_s$, producing latent representations:

$$\mathbf{z}_x = f_{c,\mathcal{X}}(\mathbf{x}) \in \mathbb{R}^{N \times D}, \quad \mathbf{z}_y = f_{c,\mathcal{Y}}(\mathbf{y}) \in \mathbb{R}^{N \times D}, \quad \mathbf{z}_s = f_{s,\mathcal{Y}}(\mathbf{y}) \in \mathbb{R}^D. \quad (1)$$

We perform the EPR and APT tasks by decoding from these latent representations:

$$\text{EPR:} \quad \hat{\mathbf{y}} = g_{\mathcal{Y}}(\mathbf{z}_x \oplus \mathbf{z}_s), \qquad \text{APT:} \quad \hat{\mathbf{x}} = g_{\mathcal{X}}(\mathbf{z}_y), \quad (2)$$

where $\oplus$ denotes broadcasted addition of the global style vector to each time step in $\mathbf{z}_x$. Both decoders are optimized via cross-entropy losses:

$$\mathcal{L}_{\text{EPR}} = \text{CE}(\hat{\mathbf{y}}, \mathbf{y}), \qquad \mathcal{L}_{\text{APT}} = \text{CE}(\hat{\mathbf{x}}, \mathbf{x}). \quad (3)$$

**Unpaired setting**    To incorporate unpaired data, we adopt a masked reconstruction objective inspired by masked autoencoders (He et al., 2022). Specifically, we define $\tilde{\mathbf{x}} = \mathrm{MASK}(\mathbf{x})$ and $\tilde{\mathbf{y}} = \mathrm{MASK}(\mathbf{y})$, where $\mathrm{MASK}(\cdot)$ randomly replaces a subset of input tokens with a special $\langle\mathrm{MASK}\rangle$ token during encoding. The model is then trained to reconstruct the full original sequence:

$$\mathcal{L}_{\mathrm{rec},\mathcal{X}} = \mathrm{CE}(g_{\mathcal{X}}(f_{c,\mathcal{X}}(\tilde{\mathbf{x}})), \mathbf{x}), \qquad \mathcal{L}_{\mathrm{rec},\mathcal{Y}} = \mathrm{CE}(g_{\mathcal{Y}}(f_{c,\mathcal{Y}}(\tilde{\mathbf{y}}) \oplus f_{s,\mathcal{Y}}(\mathbf{y})), \mathbf{y}). \qquad (4)$$

### 3.3    Latent disentanglement and regularization

We encourage disentanglement between the content space $\mathcal{Z}_c$ and the style space $\mathcal{Z}_s$ through both training objectives and architectural design. From a training perspective, The content encoders $f_{c,\mathcal{X}}(\cdot)$ and $f_{c,\mathcal{Y}}(\cdot)$ are supervised to capture score-relevant information via losses from APT, EPR, and masked reconstruction tasks. Architecturally, We represent content and style at distinct levels: $\mathbf{z}_c$ encodes fine-grained, note-level attributes such as pitch and rhythm as a sequence of latent vectors, while $\mathbf{z}_s$ summarizes the overall expressive style as a single latent vector.

To regularize the style space and promote smoothness, we impose a Kullback-Leibler divergence penalty between the posterior over $\mathbf{z}_s$ and a standard Gaussian prior:

$$\mathcal{L}_{\mathrm{KL}} = D_{\mathrm{KL}}(q(\mathbf{z}_s \mid \mathbf{y}) \,\|\, \mathcal{N}(\mathbf{0}, \mathbf{I})). \qquad (5)$$

The total training objective integrates three components: supervised losses from EPR and APT on paired data, reconstruction losses from masked inputs on unpaired data, and KL regularization on the style representation:

$$\mathcal{L}_{\mathrm{total}} = \underbrace{\mathcal{L}_{\mathrm{EPR}} + \mathcal{L}_{\mathrm{APT}}}_{\text{paired loss}} + \underbrace{\mathcal{L}_{\mathrm{rec},\mathcal{X}} + \mathcal{L}_{\mathrm{rec},\mathcal{Y}}}_{\text{unpaired loss}} + \underbrace{\mathcal{L}_{\mathrm{KL}}}_{\text{regularization}}. \qquad (6)$$

### 3.4    Modeling of performance style recommendation

After training the joint model with disentangled representations, we introduce an independent performance style recommendation (PSR) module that generates style embeddings conditioned solely on score content. This setup mimics the behavior of a pianist who selects an expressive style based on the music score alone. The goal is to model the distribution of plausible performance styles for a given score $\mathbf{x}$, enabling flexible and automated expressive rendering.

**Training**    Given a paired sample $(\mathbf{x}, \mathbf{y})$, the ground-truth style embedding $\mathbf{z}_s = f_{s,\mathcal{Y}}(\mathbf{y})$ is extracted from our frozen, pre-trained joint model. A separate score encoder $f_{g,\mathcal{X}}(\cdot)$ concurrently extracts a global content representation $\mathbf{e}_g = f_{g,\mathcal{X}}(\mathbf{x})$. We then adopt a denoising diffusion probabilistic model (DDPM) (Ho et al., 2020) to learn the conditional distribution $p(\mathbf{z}_s \mid \mathbf{e}_g)$, jointly training the diffusion denoiser and $f_{g,\mathcal{X}}(\cdot)$. The forward process perturbs the style vector by adding Gaussian noise:

$$\mathbf{z}_s^t = \sqrt{\bar{\alpha}_t}\, \mathbf{z}_s + \sqrt{1 - \bar{\alpha}_t}\, \boldsymbol{\epsilon}, \quad \boldsymbol{\epsilon} \sim \mathcal{N}(\mathbf{0}, \mathbf{I}), \qquad (7)$$

and the reverse process learns to denoise $\mathbf{z}_s^t$ conditioned on $\mathbf{e}_g$ and the diffusion step $t$. The style generator $g_s(\cdot)$ is trained to predict the added noise and is optimized using the following objective:

$$\mathcal{L}_{\mathrm{PSR}} = \mathbb{E}_{\mathbf{z}_s, \mathbf{e}_g, t, \boldsymbol{\epsilon}} \left[ \left\| \boldsymbol{\epsilon} - g_s(\mathbf{e}_g, \mathbf{z}_s^t, t) \right\|_2^2 \right]. \qquad (8)$$

**Inference**    At inference time, given $\mathbf{x}$, a style embedding $\hat{\mathbf{z}}_s$ is generated by sampling from a standard Gaussian prior and iteratively denoising it using the trained model, conditioned on $\mathbf{e}_g = f_{g,\mathcal{X}}(\mathbf{x})$. The resulting pair $(\mathbf{x}, \hat{\mathbf{z}}_s)$ is passed to the decoder $g_{\mathcal{Y}}(\cdot)$ to synthesize the expressive performance $\hat{\mathbf{y}}$.

### 3.5    Model architecture

**Joint model of EPR and APT**    As illustrated in Figure 1, the joint model consists of five transformer-based components: Score Encoder, Performance Encoder, Style Encoder, Score Decoder, and Performance Decoder. Each component adopts a standard transformer architecture (Vaswani et al., 2017) with six layers and eight attention heads, selected for their ability to model long-range dependencies and scale effectively to large symbolic music datasets. We employ rotary positional encodings (Su et al., 2024), pre-layer normalization (Brown et al., 2020), and SwiGLU activations (Shazeer, 2020), with a feed-forward hidden dimension of 3072. Decoder outputs are projected to token distributions via parallel linear layers where applicable. To obtain a global style embedding, we follow the BERT architecture (Devlin et al., 2019) in the Style Encoder by prepending a special $\langle\mathrm{CLS}\rangle$ token to the input sequence and taking the final hidden state corresponding to this token as the style vector.

**Performance style recommendation**  A separate transformer encoder, architecturally aligned with the Style Encoder, is used to extract a global score representation. A ⟨CLS⟩ token is prepended to the input score sequence, and its final hidden state is used as the global content embedding $\mathbf{e}_g$, which conditions the style generation process.

During training, a ground-truth style vector $\mathbf{z}_s$, obtained from the joint model, is perturbed using a forward diffusion process. The diffusion timestep $t$ is encoded using sinusoidal positional embeddings and concatenated with $\mathbf{e}_g$ and the noisy style vector $\mathbf{z}_s^t$. This combined representation is passed through a feed-forward network (FFN) to predict the injected noise $\boldsymbol{\epsilon}$. The model is trained using a mean squared error (MSE) loss between the predicted and true noise.

## 4 EXPERIMENTS

### 4.1 DATASETS

We use the **ASAP dataset** (Foscarin et al., 2020) for both paired training and evaluation, as it provides aligned annotations between musical scores and expressive performances. We select 967 high-quality performances and split them into training, validation, and test sets with an 8:1:1 ratio, same as Beyer & Dai (2024). To enable unpaired training, we curate an **unpaired score dataset** consisting of 75,913 public-domain MusicXML files collected from MuseScore[1]. We also compile an **unpaired performance dataset** by sourcing piano cover videos from YouTube and transcribing the audio into performance MIDI using a state-of-the-art audio-to-MIDI transcription model[2]. The model is selected based on a pilot study demonstrating strong accuracy in both note and pedal transcription. To evaluate the generalization of disentangled representations in out-of-distribution (*OOD*) settings, we additionally use the **ATEPP dataset** (Zhang et al., 2022), which contains 11,674 performances by 49 pianists spanning 25 composers, with explicit annotations of both composer and performer identities.

### 4.2 TRAINING SETUP

The joint model is trained on 3 NVIDIA A5000 GPUs with a total batch size of 36 sequences, each containing 256 notes. Optimization is performed using AdamW (Loshchilov & Hutter, 2019) for 40,000 steps, with a cosine decay learning rate schedule and linear warmup over the first 4,000 steps, peaking at $5 \times 10^{-5}$. Following the regularization strategy proposed by Beyer & Dai (2024), we apply input dropout (masking) to the decoder sequences to prevent the model from relying solely on local history and to encourage the learning of long-range structural dependencies. Specifically, we use a mask rate of 0.75 for the score decoder in the APT task and 0.2 for the MIDI decoder in the EPR task. The PSR model is trained separately on a single GPU with a batch size of 48, using the same schedule but with a peak learning rate of $1 \times 10^{-4}$.

### 4.3 METRICS

**APT**  We evaluate APT using two widely adopted metrics: **MUSTER** (Nakamura et al., 2018; Hiramatsu et al., 2021) and **ScoreSimilarity** (Suzuki, 2021; Cogliati & Duan, 2017). MUSTER assesses high-level transcription accuracy with a focus on rhythmic structure, including sub-metrics such as pitch edit distance ($E_\text{p}$), missing notes ($E_\text{miss}$), extra notes ($E_\text{extra}$), onset deviation ($E_\text{onset}$), and offset deviation ($E_\text{offset}$). ScoreSimilarity also captures pitch-level edit distances ($E_\text{miss}$, $E_\text{extra}$), with additional metrics for stem direction ($E_\text{stem}$), pitch spelling ($E_\text{spell}$), and staff assignment ($E_\text{staff}$).

**EPR**  We use both objective and subjective evaluations. **Objectively**, we compare the generated performance to its human reference and compute three metrics: *alignment rate*, *insertion rate*, and *missing rate*. Additionally, we conduct objective statistics using three metrics (Tang et al., 2023; Zhang et al., 2024): per-note variance of onset, duration, and velocity; KL divergence from human distributions; and note-aligned mean absolute error (MAE) relative to human references. **Subjectively**, we conduct a listening test with eleven participants trained in music performance. We randomly sample five pieces from Bach, Rachmaninoff, Schubert, Scriabin, and Ravel to cover a range of

---

[1] https://musescore.com/
[2] https://github.com/EleutherAI/aria-amt

Table 1: APT results on the ASAP dataset. Lower values indicate better performance across all metrics. The best results are shown in **bold**, and the second-best are underlined. Statistical significance with respect to the end-to-end baseline is denoted by † for $p < 0.05$ and ‡ for $p < 0.01$.

| Method | MUSTER | | | | | | ScoreSimilarity | | | | | |
|---|---|---|---|---|---|---|---|---|---|---|---|---|
| | $E_{\text{p}}$ | $E_{\text{miss}}$ | $E_{\text{extra}}$ | $E_{\text{onset}}$ | $E_{\text{offset}}$ | $E_{\text{avg}}$ | $E_{\text{miss}}$ | $E_{\text{extra}}$ | $E_{\text{dur.}}$ | $E_{\text{staff}}$ | $E_{\text{stem}}$ | $E_{\text{spell}}$ |
| Neural Liu et al. (2022) | **2.02** | 6.81 | 9.01 | 68.28 | 54.11 | 28.04 | 17.10 | 17.67 | 66.98 | **6.86** | – | 9.71 |
| MuseScore MuseScore (2002) | 2.41 | 7.35 | 9.64 | 47.90 | 49.44 | 23.35 | 16.17 | 16.74 | 55.23 | 21.87 | 29.87 | 9.69 |
| Finale MakeMusic, Inc. (1988) | 2.47 | 10.10 | 13.46 | 31.85 | 45.34 | 20.64 | 14.72 | 16.43 | 53.35 | 21.79 | **26.74** | 15.34 |
| Shibata et al. (2021) (J-Pop) | 2.09 | **6.38** | 8.67 | 25.02 | 29.21 | 14.27 | 10.80 | 11.39 | 71.38 | – | – | – |
| Shibata et al. (2021) (Classical) | 2.11 | 6.47 | 8.75 | 22.58 | 29.84 | 13.95 | 10.74 | 11.28 | 64.73 | – | – | – |
| End-to-end Beyer & Dai (2024) | 2.73 | 8.40 | 8.95 | 17.48 | 32.92 | 14.10 | 12.89 | 11.29 | 55.04 | 11.32 | 30.51 | 14.31 |
| Ours | 3.08‡ | 8.43 | **7.33**‡ | **16.26**† | 27.30‡ | **12.48**‡ | 13.43 | **9.48**‡ | 51.75 | 9.43‡ | 28.60 | **6.24**‡ |

Table 2: Objective evaluation of EPR results. We compare variance ($\sigma^2$), KL divergence, and MAE for onsets ($O$), durations ($D$), and velocities ($V$). For $\sigma^2$, values closer to the *Human* reference are better. For all other metrics, lower is better. Best results are in **bold**; second-best are underlined. Different letters within a column indicate statistically significant differences ($p < 0.01$).

| Method | $\sigma^2$ ($O$) | $\sigma^2$ ($D$) | $\sigma^2$ ($V$) | KL ($D$) | MAE ($D$) | KL ($V$) | MAE ($V$) |
|---|---|---|---|---|---|---|---|
| Human | $0.12^a$ | $1.72^a$ | $241.04^a$ | – | – | – | – |
| Score | $0.07^a$ | $0.07^b$ | $1.36^b$ | $13.01^a$ | $0.46^{ab}$ | $13.00^a$ | $29.14^a$ |
| DExter Zhang et al. (2024) | $0.20^b$ | $4.15^c$ | $238.86^a$ | **1.48**$^b$ | $0.88^c$ | $2.32^b$ | $24.27^b$ |
| VirtuosoNet Jeong et al. (2019) | $0.02^c$ | $0.03^d$ | $52.54^c$ | $5.72^{cd}$ | $0.48^a$ | $4.91^c$ | $14.40^c$ |
| EPR-Only | $0.03^c$ | $0.67^e$ | $126.04^d$ | $6.43^c$ | $0.42^d$ | $2.05^b$ | $10.65^d$ |
| **Ours (Target)** | $0.02^c$ | $0.58^f$ | $151.03^e$ | $5.51^d$ | **0.37**$^e$ | $1.76^d$ | $10.33^d$ |
| **Ours (PSR)** | $0.02^c$ | $0.33^e$ | $161.51^f$ | $6.19^c$ | $0.44^b$ | $2.67^e$ | $15.24^e$ |

Table 3: Objective evaluation of EPR accuracy on test samples using alignment (Align), insertion (Insert), and missing (Miss) rates ($p < 0.01$).

| Method | Align ↑ | Insert ↓ | Miss ↓ |
|---|---|---|---|
| Score | $93.52^a$ | $3.57^a$ | $2.91^a$ |
| DExter Zhang et al. (2024) | $91.27^b$ | $5.11^b$ | **3.62**$^b$ |
| VirtuosoNet Jeong et al. (2019) | $91.88^c$ | $4.23^a$ | $3.90^c$ |
| **Ours (Target)** | $91.55^d$ | $4.13^b$ | $4.32^d$ |
| **Ours (PSR)** | **92.27**$^a$ | $3.77^c$ | $3.96^a$ |

Table 4: Performer (Perf) and composer (Comp) identification results based on performance style (Style) and score content (Cont).

| Setting | F1 | Recall | Precision | Acc. |
|---|---|---|---|---|
| Style→Perf | **25.82** | **25.67** | **27.80** | **42.07** |
| Cont→Perf | 0.74 | 2.02 | 0.46 | 9.94 |
| Style→Comp | **52.45** | **50.29** | **55.99** | **77.46** |
| Cont→Comp | 3.03 | 4.66 | 3.75 | 29.99 |

genres and styles. Each participant rates the outputs in randomized order on a 5-point Likert scale (1–5) across four dimensions: *dynamics*, *tempo*, *style*, and *overall human-likeness*.

## 5 RESULTS

### 5.1 EPR AND APT PERFORMANCE

**APT** In Table 1, we present the APT performance of our model and baseline systems, with statistical significance evaluated using the Wilcoxon signed-rank test (Wilcoxon, 1945). Our model achieves performance comparable to the state-of-the-art APT system, indicating that the learned score representations capture key musical attributes such as pitch, rhythm, and structure. Our alignment-free Seq2Seq formulation achieves competitive results without requiring explicit note-level alignment. In contrast, methods such as Liu et al. (2022) and Shibata et al. (2021) attain lower pitch errors by relying on note-aligned data, which simplifies pitch and onset prediction, but limits flexibility in musically complex, one-to-many contexts (e.g., ornaments, trills, or expressive deviations).

**EPR** We compare against two strong alignment-based baselines: VirtuosoNet Jeong et al. (2019) and DExter Zhang et al. (2024). Our method is evaluated under two conditions: with extracted target styles (Ours–Target) and with PSR-generated styles (Ours–PSR). To specifically examine how our joint framework influences EPR performance, we introduce an EPR-Only variant that retains only the Score Encoder, Style Encoder, and Performance Decoder (Section 3.5), and is trained solely on the ASAP dataset. We also take score MIDI (Score) as a baseline model; it is shaded in gray in Table 2 and Table 3 to indicate that it is not an EPR model and serves only as a comparison anchor. Statistical

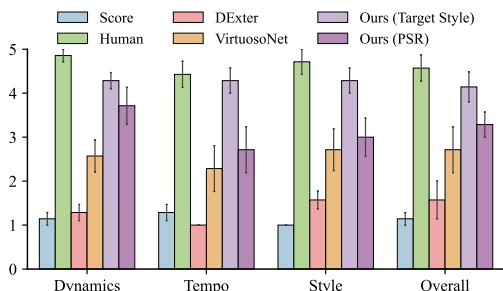 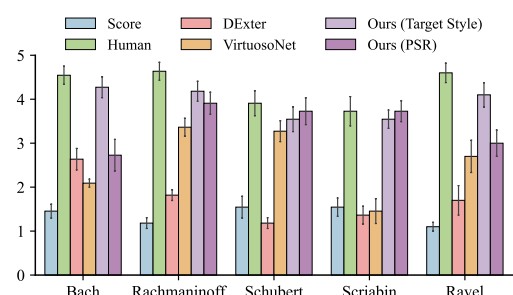

(a) Subjective ratings of PSR outputs across *musical attributes* (dynamics, tempo, and style).

(b) Breakdown of the overall subjective ratings by *composers*.

Figure 2: Subjective evaluation of expressive piano rendering performance across different systems, including human renditions, direct-from-score, baselines, and our proposed models.

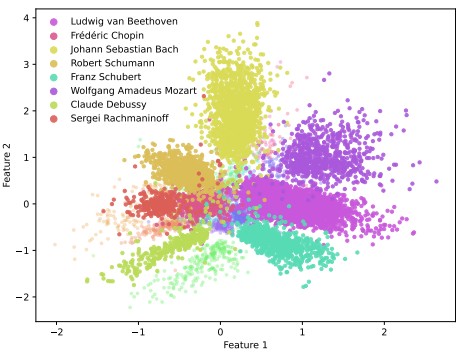 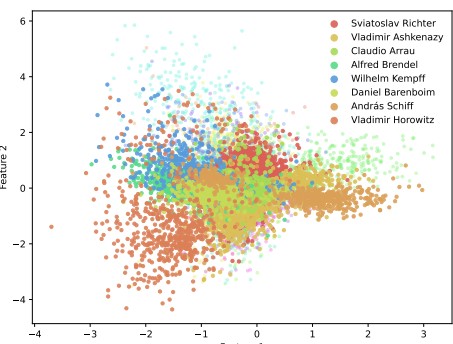

(a) Two-dimensional projection of style embeddings, colored by *composer* clusters.

(b) Two-dimensional projection of style embeddings, colored by *performer* clusters.

Figure 3: Two-dimensional visualization of performance style representations from real performances, with colors indicating clusters by composer or performer.

significance is computed by Wilcoxon signed rank test (Wilcoxon, 1945) between our methods and all baselines, with $p < 0.01$.

The objective statistics in Table 2 show that our models exhibit duration and velocity variances that more closely match those of human performances compared with other baselines, reflecting more natural expressive variability. While DExter shows even larger duration and velocity variance, this does not translate to better quality, as listening tests suggest it results from unstable dynamics rather than meaningful expressiveness. In contrast, our models achieve lower KL and MAE scores than most baselines (especially Ours–Target), confirming that they faithfully replicate the fine-grained expressive details found in human renditions. Moreover, the consistent improvement of Ours–Target over the EPR-Only variant indicates that joint modeling, together with training on additional unpaired data, leads to better EPR performance, validating the effectiveness of our joint framework.

The accuracy evaluation in Table 3 shows that Ours (PSR) achieves the highest alignment rate (92.27%) and the lowest insertion rate (3.77%), demonstrating the effectiveness of our alignment-free sequence-to-sequence formulation. Subjective results in Figure 2 show that Ours (Target) achieves the highest ratings across all attributes and styles, with Ours (PSR) closely following and outperforming baseline systems. Both variants perform strongly across composers, particularly on Bach and Scriabin.

## 5.2 REPRESENTATION DISENTANGLEMENT

**Performer/composer identification**   To further analyze the structure of the learned representations, we perform *performer and composer identification* using score content and performance style representations on the ATEPP dataset (Zhang et al., 2022), which is split into training, validation, and test sets with an 8:1:1 ratio. We evaluate four model configurations: using either the score content or performance style representation as input, and predicting either the composer or performer as the target. Each performance MIDI is segmented into 256-note chunks and processed by the trained

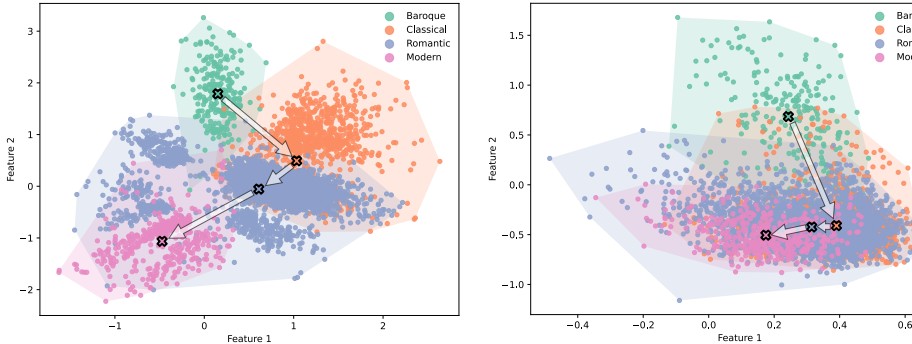

(a) Two-dimensional projection of style embeddings extracted from *actual performances* using the joint model.

(b) Two-dimensional projection of style embeddings generated by the *PSR model* from corresponding scores.

Figure 4: Two-dimensional visualization of style representations across historical eras. Colored regions denote era-specific clusters with centroids marked by black crosses; white arrows indicate temporal progression of musical styles.

joint model to extract latent representations, which are then averaged across chunks to obtain a single representation per piece. For visualization, we insert a 2D bottleneck layer before the classification head and project the resulting embeddings onto a 2D plane. The classification results and visualization are presented in Table 4 and Figure 3, respectively.

The results in Table 4 demonstrate the effectiveness of the disentangled representations. Classifiers using the style representation $\mathbf{z}_s$ achieve substantially higher composer and performer accuracy than those using the content representation $\mathbf{z}_c$, confirming successful disentanglement of performance style from score content. While $\mathbf{z}_c$ primarily encodes pitch and rhythmic structure, it is expected to preserve performance-independent musical characters (e.g., composer-specific information). This explains why the composer classifier using $\mathbf{z}_c$ (Cont→Comp) still achieves a non-trivial accuracy of 29.99%. Notably, the composer classifier using $\mathbf{z}_s$ (Style→Comp) shows much higher accuracy (77.46%). Beyond the effective disentanglement, we attribute this result to two other factors: first, as a global embedding, $\mathbf{z}_s$ is better suited for capturing high-level stylistic features than the note-level $\mathbf{z}_c$; second, professional pianists often align their performance style with the composer's stylistic conventions, thereby encoding composer information directly into their expression.

The visualization in Figure 3 further supports our findings, with style embeddings forming clear clusters by composer and performer. We also observe that embeddings from human performances contain information about both the artist and the composition. This further supports our assumption that skilled pianists adapt their style to the piece, validating the motivation behind our PSR module.

**Style transfer evaluation**   To further evaluate the disentanglement of content and style, we conducted a subjective listening test on style transfer between pieces from distinct genres. For each test case, listeners rated generated outputs on two criteria: *style similarity* to a reference performance and *overall listening quality*. We compared three conditions for the rendered style: the original (Original), the transferred reference style (Target), and an interpolation of both (Mean) to study the learned style feature space. As shown in Figure 5, the Target condition achieves the highest style similarity ratings in Samples 1 and 3, indicating successful transfer. Notably, this improvement does not compromise overall quality. The Mean condition yields consistently strong quality across all samples, suggesting that the style space is well-structured and supports smooth interpolation.

## 5.3 EFFECTIVENESS OF PSR

To evaluate the styles generated by the PSR model, we collect 5,003 performances from the ATEPP dataset with aligned scores. For each performance, we obtain two style vectors: one extracted directly from the performance using the joint model, and one generated from the corresponding score using the PSR model. Based on the title and composer metadata parsed by GPT-4o mini (Achiam et al., 2023), each piece is assigned to one of four historical eras including Baroque, Classical, Romantic, and Modern.

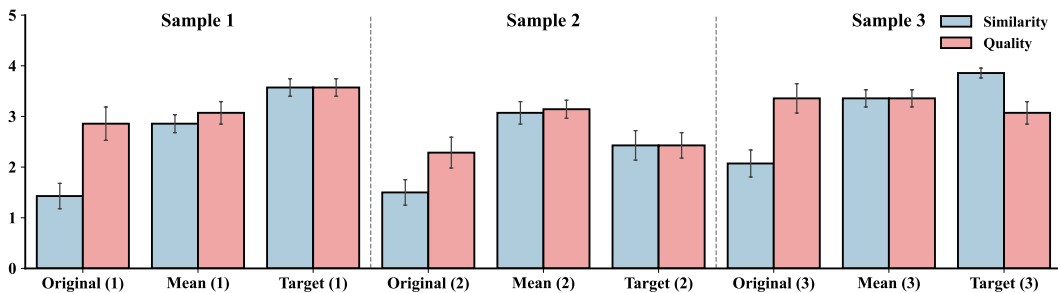

Figure 5: Subjective ratings for three generated samples using different style settings. Listeners rated each output on style similarity and overall listening quality.

We project the style vectors into 2D using the classifier from Section 5.2. As shown in Figure 4, the PSR-generated styles (right) closely mirror those extracted from real performances (left), exhibiting similar clustering structure, era-wise separation, and centroid locations. This alignment, together with the subjective results in Figure 2, supports the PSR model's ability to synthesize stylistically meaningful embeddings from score content alone.

## 6 CONCLUSION

In this paper, we present a unified framework for expressive piano performance rendering (EPR) and automatic piano transcription (APT), built upon disentangled latent representations of score content and performance style. To enable flexible style-aware rendering, we introduce a DDPM-based Performance Style Recommendation (PSR) module that generates expressive styles directly from score content. Evaluated through objective metrics, subjective listening tests, and representation visualizations, our approach achieves performance on par with state-of-the-art methods across both EPR and APT tasks. Our findings demonstrate that: (a) the joint model effectively learns disentangled representations of content and style; (b) EPR can be formulated as a sequence-to-sequence task without requiring note-level alignment; (c) the model supports flexible style transfer; and (d) the PSR module produces stylistically appropriate outputs conditioned solely on the score. As future work, we aim to extend this framework to popular music, which presents greater stylistic diversity and practical relevance than classical music.

## ETHICS STATEMENT

The authors have reviewed and conformed in every respect with the ICLR Code of Ethics `https://iclr.cc/public/CodeOfEthics`. The human study in our experiment is based on online crowdsourcing, which bears minimum risk. Participants are informed that participation in our study is enstirely voluntary and that they may choose to stop participating at any time without any negative consequences. No personally identifying information is collected in the human study.

## REPRODUCIBILITY STATEMENT

We introduce our dataset and experimental settings in Section 4.1 and ection 4.2, respectively. We also provide details of model architectures necessary for reproduction in Appendix B. The code will be released upon acceptance with sufficient instructions for reproducing the model architecture and training pipeline using public datasets such as ASAP and ATEPP.

## ACKNOWLEDGMENTS

We would like to thank Qihao Liang for his valuable advice on writing. We also appreciate Xichu Ma for his assistance during the rebuttal process.

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

APPENDICES

The appendix is structured into 7 main parts. Appendix A specifies the data processing details involved in the paper. Appendix B presents implementation details of our proposed methods. Appendix C provides subjective listening test details. Appendix D presents supplementary experimental results on GPT-4o mini output verification, diversity analysis of EPR, and ablation studies. Appendix E discusses challenges and future work. In Appendix F, we provide several examples of expressive piano rendering (EPR) and automatic piano transcription (APT). Finally, we disclose the use of LLMs in Appendix G.

# A    DATA PROCESSING DETAILS

## A.1    DATA FILTERING

To construct a clean and consistent symbolic dataset from MuseScore, we apply a series of rule-based filters to exclude low-quality or incompatible piano scores. A score is retained only if it satisfies all of the following criteria:

- **Staff structure:** The score must contain exactly two staves, conforming to standard piano notation.
- **Note count:** The total number of notes must be at least 100.
- **Bar count:** The score must span at least 10 bars.
- **Note density:** No individual bar may contain more than 64 notes, to avoid overly dense notation.
- **Time signature:** The time signature must fall within a musically plausible range: the number of beats per measure must be between 1 and 16, and the beat type must belong to the set $\{2, 4, 8, 16, 32\}$.
- **Key signature:** The notated key signature, expressed as the number of fifths, must lie within $[-7, 7]$. In addition, the mean distance between the notated and estimated keys (Temperley, 1999; Cancino-Chacón et al., 2022) must not exceed 1.

To compute key signature distance, we segment each score into contiguous regions with a constant notated key signature. For each segment, we estimate the key and compare it to the notated key. Let $k_i \in [-7, 7]$ denote the notated key signature and $\hat{k}_i \in [-7, 7]$ the estimated key. The key distance is defined as:

$$d_i = \min\left(|k_i - \hat{k}_i|,\ |k_i - \hat{k}_i + 12|,\ |k_i - \hat{k}_i - 12|\right), \tag{9}$$

accounting for circularity in the circle of fifths. The final mean key distance is computed as:

$$D = \frac{1}{N} \sum_{i=1}^{N} d_i, \tag{10}$$

where $N$ is the number of key-stable segments. Only scores with $D \leq 1$ are retained.

## A.2    DATA REPRESENTATION DETAILS

**Score**    The score representation captures structural and timing information relevant for expressive rendering. The input encodes performance-related features, while the output is extended to include additional notation-specific attributes necessary for producing readable sheet music.

Time-based features, including inter-onset interval (IOI), onset-in-bar, note value, and downbeat, are discretized into consistent vocabularies spanning 0 to 6 quarter lengths, each with 145–146 bins. Boolean-valued attributes, such as *grace note* and *hand/staff assignment*, are encoded as binary values. The score output additionally predicts symbolic formatting elements such as voice number, articulation markings (e.g., trill, staccato), and engraving-specific cues including stem direction and accidentals (e.g., double flats and sharps). All features are treated as discrete classification targets using small, well-defined vocabularies summarized in Table 5.

Table 5: Vocabulary size and value ranges of input and output parameters for music score.

| Parameter | $N_{\text{vocab}}$ | Range/Values | Input | Output |
|---|---|---|---|---|
| Onset-in-Bar | 145 | [0, 6] quarter-length | ✓ | ✓ |
| Inter-Onset Interval (IOI) | 145 | [0, 6] quarter-length | ✓ | |
| Pitch | 128 | [0, 127] | ✓ | ✓ |
| Note Value | 97 | [0, 4] quarter-length | ✓ | ✓ |
| Measure Length | 146 | $[0, 6] \cup \{false\}$ | ✓ | ✓ |
| Hand/Staff | 2 | boolean | ✓ | ✓ |
| Trill | 2 | boolean | ✓ | ✓ |
| Grace | 2 | boolean | ✓ | ✓ |
| Staccato | 2 | boolean | | ✓ |
| Voice | 8 | [1, 8] | | ✓ |
| Stem | 3 | {up, down, none} | | ✓ |
| Accidental | 6 | {♭♭, ♭, ♮, ♯, ♯♯, none} | | ✓ |

Table 6: Vocabulary size and value ranges of input and output parameters for performance MIDI.

| Parameter | $N_{\text{vocab}}$ | Range/Values | Input | Output |
|---|---|---|---|---|
| Pitch ($p_i$) | 128 | [0, 127] | ✓ | |
| IOI ($o_i$) | 200 | [0, 8] seconds | ✓ | |
| Duration ($d_i$) | 200 | [0, 8] seconds | ✓ | |
| Velocity ($v_i$) | 128 | [0, 127] | ✓ | |
| Pitch Token | 88 | [21, 108] | | ✓ |
| Duration Token | 64 | 64 discrete steps | | ✓ |
| Velocity Token | 32 | 32 velocity bins | | ✓ |
| Time-Shift Token | 65 | quantized by `beat_res` | | ✓ |
| Special Tokens | 2 | {`PAD`, `BOS`} | | ✓ |

**Performance MIDI**  The performance representation captures expressive aspects of human execution, including timing, articulation, and dynamics. At the input level, we extract four note-level features: **Pitch** (MIDI number), **IOI** (inter-onset interval in seconds), **Duration** (extended by pedal usage), and **Velocity** (loudness). IOI and Duration are quantized into 200 bins, while Velocity is coarsely grouped into 8 bins for robustness.

For output, we adopt a structured token-based representation (Huang & Yang, 2020), implemented using the `miditok` library (Fradet et al., 2021). The model generates discrete token sequences that include **Note-On**, **Duration**, **Velocity**, and **Time-Shift** events, enabling expressive sequence generation without explicit note-level alignment. Special tokens such as `BOS` (beginning of sequence) and `PAD` are also used to facilitate training and formatting. Table 6 provides the vocabulary sizes and ranges for all input and output features.

## B  IMPLEMENTATION DETAILS

### B.1  JOINT MODEL

Our joint model is implemented in PyTorch Lightning and trained via multi-task learning to simultaneously handle EPR, APT, and masked reconstruction from unpaired data. This section outlines the training tasks, loss formulation, optimization strategy, and implementation setup.

**Training tasks**  Each training step involves four supervised or self-supervised subtasks:

- **APT** The score decoder reconstructs symbolic score tokens from the performance content encoder.

- **EPR** The performance decoder generates MIDI tokens conditioned on the score content encoder and a style embedding.

- **Score Reconstruction** The score encoder is trained using random masking to reconstruct full sequences from partially masked inputs.

- **MIDI Reconstruction** The performance content encoder and decoder reconstruct MIDI sequences from masked inputs in a similar fashion.

Additionally, a Kullback-Leibler (KL) regularization term is applied to the style embedding to encourage compactness and diversity in the latent style space.

**Training loss** Let $\mathcal{L}_{\text{APT}}$, $\mathcal{L}_{\text{EPR}}$, $\mathcal{L}_{\text{rec},\mathcal{X}}$, and $\mathcal{L}_{\text{rec},\mathcal{Y}}$ denote the cross-entropy losses for APT, EPR, score reconstruction, and MIDI reconstruction, respectively. The total training objective is given by:

$$\mathcal{L}_{\text{total}} = \mathcal{L}_{\text{APT}} + \mathcal{L}_{\text{EPR}} + \lambda_{\text{rec}} \cdot (\mathcal{L}_{\text{rec},\mathcal{X}} + \mathcal{L}_{\text{rec},\mathcal{Y}}) + \lambda_{\text{KL}} \cdot \mathcal{L}_{\text{KL}}, \tag{11}$$

where $\lambda_{\text{rec}} = 0.2$ and $\lambda_{\text{KL}} = 0.1$. We apply a 50% masking rate to encoder inputs during reconstruction, and a lighter masking rate of 10–20% to decoder inputs to improve robustness and mitigate overfitting.

**Optimization** We use AdamW optimizers (Loshchilov & Hutter, 2019) with a learning rate of $5 \times 10^{-5}$, following a cosine learning rate schedule with 4,000 warm-up steps and 40,000 total steps. Gradient updates are manually scheduled, and training is performed using mixed precision (fp16).

**Batching and scheduling** Each training step processes 144 sequences (each of length 256 notes), evenly divided among the four subtask types: APT, EPR, unpaired score, and unpaired MIDI. Data loaders for each subset are interleaved and sampled in parallel. KL regularization is computed once per batch using the mean and variance of the predicted style embeddings.

**Implementation notes** All model components use a unified embedding dimension of $d = 512$, with task-specific embedding layers. Attention masks are dynamically modified during training to simulate incomplete inputs, following masked language modeling strategies. The system is trained on 3 NVIDIA A5000 GPUs using batch-level data parallelism.

### B.2 PERFORMANCE STYLE RECOMMENDATION (PSR)

The performance style recommendation (PSR) module is designed to generate expressive style embeddings directly from symbolic scores, enabling performance rendering without requiring paired expressive data at inference time. The overall architecture is illustrated in Figure 6.

**Overview** The PSR model comprises two components: (1) a transformer-based score encoder that extracts a global content embedding from a symbolic score sequence, and (2) a denoising diffusion probabilistic model (DDPM) that generates a style vector conditioned on this content embedding. This pipeline enables sampling stylistically coherent vectors from Gaussian noise, guided by the structure of the input score.

**Score encoder** We adopt a transformer encoder $f_{g,\mathcal{X}}(\mathbf{x})$ to process the input score sequence. Following the BERT-style design (Devlin et al., 2019), a special `[CLS]` token is prepended to the sequence, and its final-layer hidden state is used as the *global* score content representation $\mathbf{e}_g \in \mathbb{R}^D$.

**Diffusion network** We employ a DDPM (Ho et al., 2020) with velocity prediction (Salimans & Ho, 2022) to model the conditional distribution over style embeddings given the content vector. During training, the model learns to recover a ground-truth style vector $\mathbf{z}_s$, extracted from human performances via the joint model, from a noisy version $\mathbf{z}_s^t$ produced by the forward diffusion process. A sinusoidal timestep embedding $\mathbf{e}_t$ is concatenated with the projected content embedding $\mathbf{e}_g'$ and the noisy style vector $\mathbf{z}_s^t$, and passed through a multi-layer perceptron (MLP) to predict the velocity target $\mathbf{v}_{\text{target}}$. The model is optimized with the following mean squared error loss:

$$\mathcal{L}_{\text{PSR}} = \mathbb{E}_{\mathbf{z}_s, \mathbf{e}_g, t} \left[ \left\| g_s(\mathbf{z}_s^t, \mathbf{e}_t, \mathbf{e}_g') - \mathbf{v}_{\text{target}} \right\|_2^2 \right]. \tag{12}$$

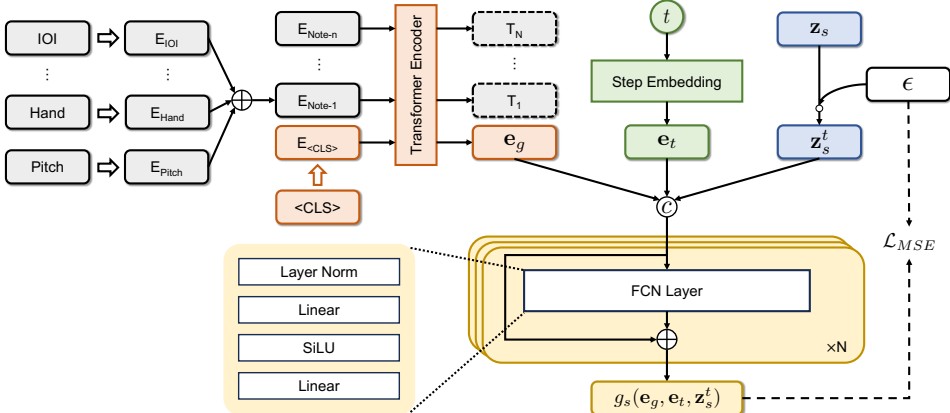

Figure 6: Architecture of the performance style recommendation (PSR) module. Given a symbolic score, we extract a global content embedding using a transformer encoder and train a diffusion model to predict the style embedding from noise.

Table 7: Comparison of model parameters and inference speed for APT and EPR.

| Method | Number of Parameters | APT Inference speed | EPR Inference speed |
|---|---|---|---|
| Dexter (Zhang et al., 2024) | 62.41 M | - | 14.14 s/sample |
| Virtuosonet (Jeong et al., 2019) | 5.16 M | - | 0.35 s/sample |
| End-to-End (Beyer & Dai, 2024) | 32.60 M | 4.56 s/sample | - |
| Ours | 188.21 M | 4.86 s/sample | 10.17 s/sample |

**Inference**    At inference time, a style vector is initialized from a standard Gaussian distribution and iteratively denoised using the exponential moving average (EMA) version of the MLP denoising network. The resulting style embedding $\hat{\mathbf{z}}_s$ can be combined with the score content to condition the expressive rendering model. This one-to-many mapping enables diverse, plausible, and stylistically appropriate generation from symbolic input alone.

### B.3 MODEL COMPLEXITY

Table 7 summarizes the model sizes and inference speeds for both APT and EPR, tested on a single NVIDIA A5000 GPU. Several observations can be drawn. First, although our unified model contains substantially more parameters (188.21M) than the end-to-end APT baseline (32.60M), its APT inference speed (4.86s/sample) remains comparable because only the APT-specific modules are active during APT decoding; the additional EPR-related parameters are not involved in this forward pass. Second, APT inference is consistently faster than EPR within the unified architecture. This follows from the shorter output sequences in APT, whereas EPR must generate longer sequences under the structured performance representation (Section 3.1). Third, EPR inference speed varies widely across baseline systems. VirtuosoNet is the fastest (0.35s/sample) as it is trained on note-aligned data and does not require autoregressive decoding, while DExter is the slowest (14.14s/sample) due to its diffusion-based design requiring multiple denoising iterations. Our unified model falls between these two extremes: despite performing autoregressive decoding and modeling fine-grained expressive attributes, it achieves a reasonable EPR inference time (10.17s/sample) while supporting both tasks within a single architecture.

## C SUBJECTIVE LISTENING TEST INSTRUCTIONS

### C.1 OVERVIEW

We conduct our subjective evaluation using a Google Form [3], structured into two sections: (1) evaluation of performance style recommendation (PSR), and (2) evaluation of style transfer. Each

---

[3] https://docs.google.com/forms

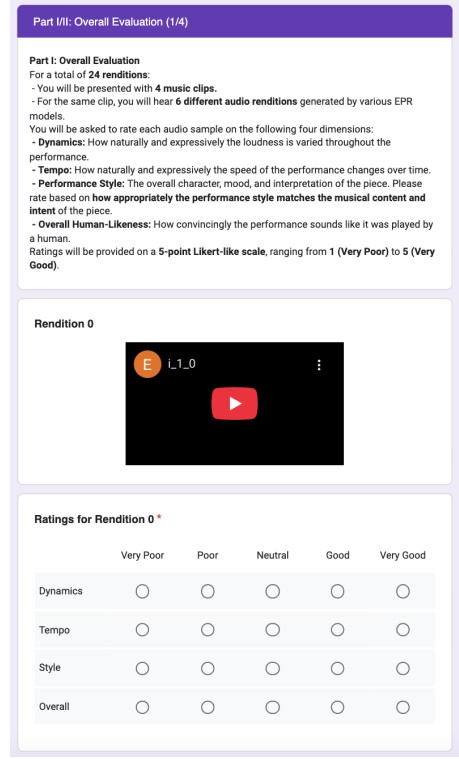

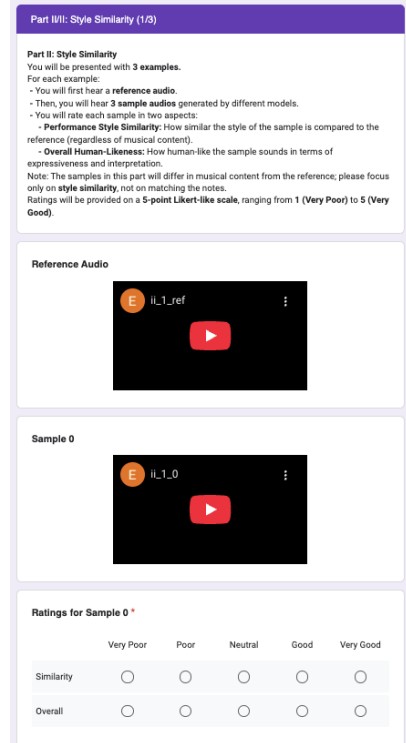

(a) Overall evaluation of EPR.      (b) Style transfer evaluation.

Figure 7: Screenshots of survey pages and instructions of our online survey.

participant completes both sections, with an average completion time of approximately 32 minutes. Figure 7 shows sample survey pages along with participant instructions. Detailed descriptions of the survey structure are provided below.

## C.2 SURVEY STRUCTURE

**Part I: Overall Evaluation** Participants are presented with 4 music clips, each accompanied by 6 audio renditions generated by different EPR models. Each rendition is rated along the following four dimensions:

- **Dynamics:** Naturalness and expressiveness of loudness variation.
- **Tempo:** Naturalness and expressiveness of tempo fluctuations over time.
- **Performance Style:** Appropriateness of the performance's character, mood, and interpretation.
- **Overall Human-Likeness:** How convincingly the performance resembles that of a human.

Ratings are provided on a 5-point Likert scale ranging from 1 (Very Poor) to 5 (Very Good).

**Part II: Style Similarity** Participants are presented with 3 examples. Each example consists of:

- A reference performance.
- Three test renditions generated by different models, with varied content but intended to share the same performance style.

Each test rendition is rated on:

- **Performance Style Similarity:** The extent to which the style (e.g., rhythm, dynamics, pedal usage) matches the reference, independent of pitch content.

Table 8: Agreement matrices between human annotators and GPT-4o mini. Cohen's $\kappa$ values: Annotator 1 (A1) v.s. Annotator 2 (A2) = *0.89*; Annotator 1 (A1) v.s. GPT-4o mini = *0.85*; Annotator 2 (A2) v.s. GPT-4o mini = *0.89*. B = Baroque, C = Classical, R = Romantic, M = Modern.

| | A1 vs. A2 | | | | A1 vs. GPT-4o mini | | | | A2 vs. GPT-4o mini | | | |
|---|---|---|---|---|---|---|---|---|---|---|---|---|
| | **B** | **C** | **R** | **M** | **B** | **C** | **R** | **M** | **B** | **C** | **R** | **M** |
| **B** | 22 | 0 | 0 | 0 | 22 | 0 | 0 | 0 | 22 | 0 | 0 | 0 |
| **C** | 0 | 10 | 4 | 0 | 0 | 6 | 8 | 0 | 0 | 6 | 4 | 0 |
| **R** | 0 | 0 | 53 | 2 | 0 | 0 | 55 | 0 | 0 | 0 | 58 | 0 |
| **M** | 0 | 0 | 1 | 8 | 0 | 0 | 1 | 8 | 0 | 0 | 2 | 8 |

Table 9: Average pairwise MAEs for human renditions and model outputs.

| | Duration MAE | Velocity MAE |
|---|---|---|
| Human | 0.06 | 11.62 |
| Model | 0.08 | 8.01 |

- **Overall Human-Likeness:** Perceived expressiveness and realism of the performance.

All ratings are again provided on a 5-point Likert scale.

### C.3 Additional notes

- Participants are instructed to evaluate variation and human-likeness, rather than personal preference or audio fidelity.

- All audio sources are anonymized; both the order of clips and model outputs are randomized to reduce potential bias.

- Participants are encouraged to use headphones in a quiet environment for optimal listening conditions.

- The total duration of the survey is approximately 25–30 minutes. No personal data is collected.

## D Supplementary experimental results

### D.1 Human verification of GPT-4o mini outputs

To assess the reliability of GPT-4o mini predictions in Section 5.3, we conducted a human verification study on 100 randomly sampled movements, independently annotated by two professionally trained pianists into four eras (Baroque, Classical, Romantic, Modern). Agreement was measured using Cohen's $\kappa = \frac{p_o - p_e}{1 - p_e}$, where $p_o$ is the observed agreement and $p_e$ is the expected agreement by chance. As shown in Table 8, inter-annotator agreement was high ($\kappa = 0.89$), and GPT-4o mini showed similarly strong consistency with both annotators ($\kappa = 0.85$ and $\kappa = 0.89$). Most disagreements occurred in transitional works between Classical and Romantic eras, where stylistic boundaries are ambiguous. For example, *Piano Sonata No. 26 in E-flat, Op. 81a "Les adieux": II. Abwesenheit (Andante espressivo)* was annotated as Classical by both human experts but labeled as Romantic by GPT-4o mini. Such cases are reasonable given the transitional nature of the repertoire. Overall, these results confirm that GPT-4o mini aligns closely with expert judgment and can be used as a reliable reference for PSR evaluation.

### D.2 Diversity analysis of EPR

To verify that the model captures one-to-many expressive variation rather than collapsing to an averaged output, we analyzed diversity on a score from ASAP with 7 human performances and 7 model outputs generated via top-$k$ sampling ($k = 5$). Pairwise note-aligned MAEs were computed

Table 10: Pairwise MAEs among 7 *human* renditions.

(a) Durations

|    | H1   | H2   | H3   | H4   | H5   | H6   | H7   |
|----|------|------|------|------|------|------|------|
| H1 | 0.00 | 0.06 | 0.07 | 0.06 | 0.06 | 0.07 | 0.06 |
| H2 |      | 0.00 | 0.06 | 0.06 | 0.05 | 0.06 | 0.05 |
| H3 |      |      | 0.00 | 0.07 | 0.07 | 0.06 | 0.06 |
| H4 |      |      |      | 0.00 | 0.06 | 0.08 | 0.05 |
| H5 |      |      |      |      | 0.00 | 0.06 | 0.05 |
| H6 |      |      |      |      |      | 0.00 | 0.06 |
| H7 |      |      |      |      |      |      | 0.00 |

(b) Velocities

|    | H1   | H2    | H3    | H4    | H5    | H6    | H7    |
|----|------|-------|-------|-------|-------|-------|-------|
| H1 | 0.00 | 10.66 | 14.11 | 15.82 | 10.94 | 12.46 | 12.90 |
| H2 |      | 0.00  | 13.23 | 13.82 | 11.01 | 12.23 | 12.26 |
| H3 |      |       | 0.00  | 9.05  | 11.42 | 9.02  | 9.33  |
| H4 |      |       |       | 0.00  | 12.16 | 10.80 | 11.26 |
| H5 |      |       |       |       | 0.00  | 11.12 | 10.78 |
| H6 |      |       |       |       |       | 0.00  | 9.66  |
| H7 |      |       |       |       |       |       | 0.00  |

Table 11: Pairwise MAEs among 7 *model* outputs.

(a) Durations

|    | M1   | M2   | M3   | M4   | M5   | M6   | M7   |
|----|------|------|------|------|------|------|------|
| M1 | 0.00 | 0.08 | 0.11 | 0.13 | 0.06 | 0.10 | 0.12 |
| M2 |      | 0.00 | 0.08 | 0.10 | 0.06 | 0.09 | 0.09 |
| M3 |      |      | 0.00 | 0.08 | 0.06 | 0.05 | 0.04 |
| M4 |      |      |      | 0.00 | 0.07 | 0.09 | 0.08 |
| M5 |      |      |      |      | 0.00 | 0.07 | 0.05 |
| M6 |      |      |      |      |      | 0.00 | 0.06 |
| M7 |      |      |      |      |      |      | 0.00 |

(b) Velocities

|    | M1   | M2   | M3    | M4   | M5    | M6   | M7   |
|----|------|------|-------|------|-------|------|------|
| M1 | 0.00 | 6.09 | 10.06 | 9.47 | 7.73  | 8.26 | 8.14 |
| M2 |      | 0.00 | 9.82  | 8.53 | 8.61  | 9.94 | 9.62 |
| M3 |      |      | 0.00  | 6.17 | 10.12 | 7.48 | 8.45 |
| M4 |      |      |       | 0.00 | 8.19  | 7.16 | 8.40 |
| M5 |      |      |       |      | 0.00  | 6.50 | 5.00 |
| M6 |      |      |       |      |       | 0.00 | 4.37 |
| M7 |      |      |       |      |       |      | 0.00 |

Table 12: APT results on different proportions of paired/unpaired data. Lower is better for all metrics. The best results are shown in **bold**, and the second-best are underlined.

| Method | MUSTER | | | | | | ScoreSimilarity | | | | | |
|--------|--------|----------|-----------|-----------|------------|---------|--------|-----------|------------|-----------|------------|-----------|
|        | $E_p$  | $E_{miss}$ | $E_{extra}$ | $E_{onset}$ | $E_{offset}$ | $E_{avg}$ | $E_{miss}$ | $E_{extra}$ | $E_{dur.}$ | $E_{staff}$ | $E_{stem}$ | $E_{spell}$ |
| paired + 0% unpaired   | 3.10 | 9.33 | 8.09 | 16.69 | 29.29 | 13.30 | 13.98 | 10.13 | 59.45 | 10.02 | 30.60 | 8.44 |
| paired + 25% unpaired  | **2.94** | 8.86 | 7.80 | 16.37 | 28.36 | 12.87 | 13.66 | 10.10 | 60.06 | 8.86 | 30.58 | 7.46 |
| paired + 50% unpaired  | 3.24 | 9.74 | 7.59 | 17.07 | 27.99 | 13.13 | 14.91 | 9.96 | 56.86 | **7.91** | 31.61 | 10.49 |
| paired + 100% unpaired | 3.08 | **8.43** | **7.33** | **16.26** | **27.30** | **12.48** | **13.43** | **9.48** | **51.75** | 9.43 | **28.60** | **6.24** |

Table 13: Performer (Perf) and composer (Comp) identification under two data settings: paired + 0% unpaired and paired + 100% unpaired. **Boldface** is kept *only* for Style→Perf and Style→Comp to highlight the effect of adding unpaired data. The rightmost block reports the per-metric gain $\Delta$ (100% unpaired − 0% unpaired).

| Setting | paired + 0% unpaired | | | | paired + 100% unpaired | | | | $\Delta$ (100% − 0%) | | | |
|---------|------|--------|-----------|------|------|--------|-----------|------|--------|--------|--------|--------|
|         | F1   | Recall | Precision | Acc. | F1   | Recall | Precision | Acc. | $\Delta$F1 | $\Delta$Rec. | $\Delta$Prec. | $\Delta$Acc. |
| Style→Perf | 19.33 | 19.17 | 20.21 | 33.76 | **25.82** | **25.67** | **27.80** | **42.07** | +6.49 | +6.50 | +7.59 | +8.31 |
| Cont→Perf  | 0.71  | 1.94  | 0.44  | 9.68  | 0.74  | 2.02  | 0.46  | 9.94  | +0.03 | +0.08 | +0.02 | +0.26 |
| Style→Comp | 46.33 | 43.51 | 55.24 | 69.07 | **52.45** | **50.29** | **55.99** | **77.46** | +6.12 | +6.78 | +0.75 | +8.39 |
| Cont→Comp  | 2.92  | 4.57  | 4.37  | 30.16 | 3.03  | 4.66  | 3.75  | 29.99 | +0.11 | +0.09 | −0.62 | −0.17 |

for durations and velocities. As summarized in Table 9, the average human MAEs were 0.06 (duration) and 11.62 (velocity), while the model achieved 0.08 and 8.01, respectively. Detailed pairwise matrices (Table 10, Table 11) show that model outputs exhibit meaningful internal variation, following the diversity observed in human renditions. This demonstrates that the proposed model captures distributional expressiveness in performance generation rather than regressing to a mean output.

### D.3 ABLATION STUDIES

**Ablations on unpaired data** To evaluate the impact of unpaired data, we conduct an ablation study by varying the ratio of unpaired data used in training. We train four model variants using 0% (paired data only), 25%, 50%, and 100% of our curated unpaired datasets, while keeping all

Table 14: Ablation of KL weight on KL divergence, active units (AU), and classification accuracy (CA).

| KL weight | KL divergence | AU | CA |
|---|---|---|---|
| 0 | 1.11 | 512 | 0.94 |
| 0.5 | 0.69 | 512 | 0.91 |
| 1 | 0.09 | 512 | 0.88 |
| 5 | 0.10 | 512 | 0.76 |

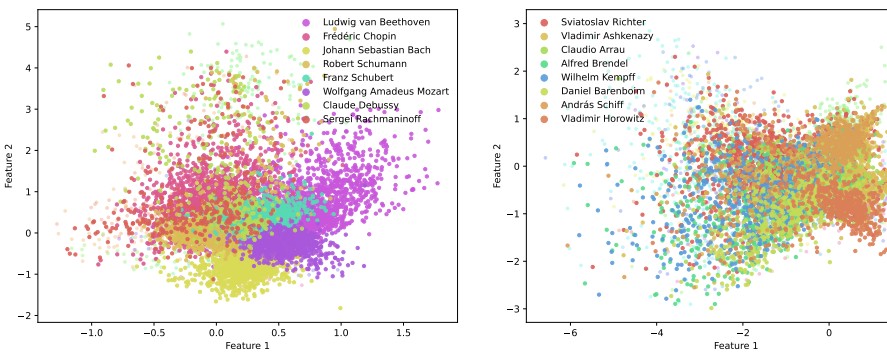

(a) Two-dimensional projection of style embeddings, colored by *composer* clusters.

(b) Two-dimensional projection of style embeddings, colored by *performer* clusters.

Figure 8: Two-dimensional performance style visualization of *paired + 0% unpaired* variant representations, with colors indicating clusters by composer or performer.

other hyperparameters constant. The APT results in Table 12 show that incorporating unpaired data generally enhances performance. Adding just 25% of the unpaired data provides a consistent improvement over the baseline model trained only on paired data, while using the full 100% unpaired dataset achieves the best overall performance.

Furthermore, to study the influence of unpaired data on representation disentanglement, we conduct *performer and composer identification* in Section 5.2. As shown in Table 13, introducing unpaired data significantly enhances the quality of the style representation. For both performer (Style→Perf) and composer (Style→Comp) identification, all metrics see a substantial improvement, with classification accuracy increasing by +8.31% and +8.39%, respectively. In contrast, the classification performance using the content representation remains almost unchanged. These results indicate that our model effectively leverages unpaired data to enrich the style embedding while successfully preserving the disentanglement between performance style and score content.

To further examine the effect of unpaired data, we visualize the learned style embeddings for the *paired + 0% unpaired* variant and the *paired + 100% unpaired* variant (Figure 8). Compared with Figure 3, the *paired + 0% unpaired* variant exhibits noticeably less structure, with composer and performer clusters partially overlapping. Incorporating unpaired data produces clearer and more compact clusters, indicating that the additional data helps the model learn more discriminative and coherent style representations.

**KL divergence analysis** We evaluate latent informativeness across different KL weights for the KL divergence loss introduced in Section 3.3 using three metrics (Wang et al., 2021): (i) KL divergence between posterior and prior, (ii) Active Units (AU) measuring the number of latent dimensions with sample variance $> 0.01$, and (iii) style classification accuracy (CA) using $\mathbf{z}_s$ and ground-truth era labels from Section 5.3. As shown in Table 14, stronger KL regularization reduces both KL divergence and classification accuracy, while the number of active units remains consistently high (512). This indicates that although some information compression occurs, the latent representation does not undergo full posterior collapse, and still preserves musically meaningful information.

## E    DISCUSSION ON CHALLENGES AND FUTURE WORK

**Beyond classical genres**    Our joint framework is inherently genre-agnostic: it does not rely on classical-specific score structures and can, in principle, generalize to any musical style given suitable supervision. Our current study focuses on classical piano performance primarily due to data availability, as existing score–performance aligned datasets such as ASAP (Foscarin et al., 2020) contain exclusively classical repertoire. Extending expressive performance rendering to other genres (e.g., jazz or pop) introduces additional challenges. First, these genres lack large curated paired datasets, making supervised learning more difficult. Second, many non-classical traditions involve improvisation, flexible phrase structures, and rhythmically nuanced conventions (e.g., swing timing) that are not well captured by classical-style fully notated scores. For instance, jazz performances are typically aligned to lead sheets rather than detailed five-line staff notation, making precise score–performance alignment inherently ambiguous. As future work, we aim to curate genre-specific datasets and adopt notation formats appropriate to each style (e.g., lead sheets for jazz, chord charts for pop) to extend our unified EPR framework beyond classical music and enable genre-aware expressive rendering.

**Transcription biases**    Our use of unpaired YouTube performances offers valuable stylistic diversity and follows a data construction strategy similar to ATEPP (Zhang et al., 2022). Nonetheless, this pipeline may also introduce transcription-related artifacts, as the audio-to-MIDI system can impose quantization biases or systematic timing regularities. As a result, the style encoder may inadvertently learn these artifacts rather than capturing purely human expressive behavior. In the long term, we aim to pursue end-to-end performance modeling by generating audio directly from score notation, thereby mitigating domain shifts introduced by intermediate MIDI representations and allowing the model to learn stylistic nuances more faithfully from raw human performances.

## F    EXAMPLES OF EPR AND APT

**EPR**    Demos are available at `https://wei-zeng98.github.io/joint-apt-epr/`. The page includes two sections: (1) rendering results from various models, including ours, on five music pieces from different composers; and (2) style transfer results on three music pieces, showcasing the flexibility of our method.

**APT**    Examples of APT outputs are shown in Figure 9–Figure 11. For each sample, the ground-truth score is displayed at the top, and the predicted score from our model at the bottom. Missing notes in the target scores are highlighted with red bounding boxes, while inserted notes in the predicted scores are highlighted with blue bounding boxes.

## G    THE USE OF LARGE LANGUAGE MODELS (LLMS)

In accordance with the ICLR policy, we disclose the use of Large Language Models (LLMs) as assistive tools in the preparation of this manuscript. The specific applications are detailed below:

- Data annotation: We employed an LLM to assist in the annotation of our dataset. The detailed methodology and human verification have been introduced in Section 5.3 and Appendix D.1.
- Literature search: LLMs were used as a tool to aid in the initial search and summarization of relevant prior work.
- Writing and polishing: We utilized an LLM for proofreading and language refinement.

All authors have carefully reviewed and edited the manuscript. We take full responsibility for all content of this paper, including the final research ideas, experimental results, and the accuracy and integrity of the text.

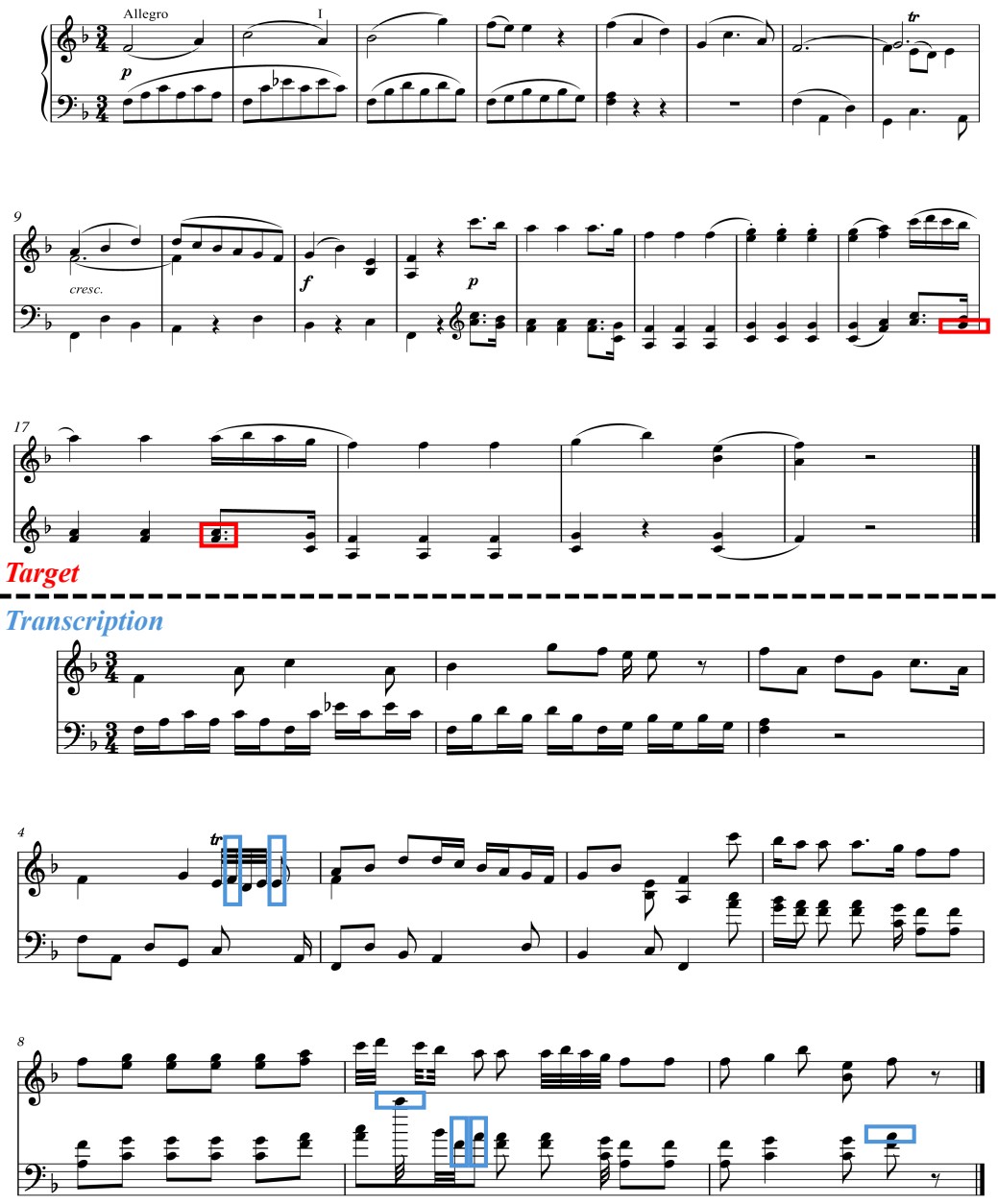

Figure 9: Ground truth score (*upper*) and transcribed score (*lower*) from *Piano Sonata No.12 in F Major, K 332*, by Wolfgang Amadeus Mozart (APT sample 1).

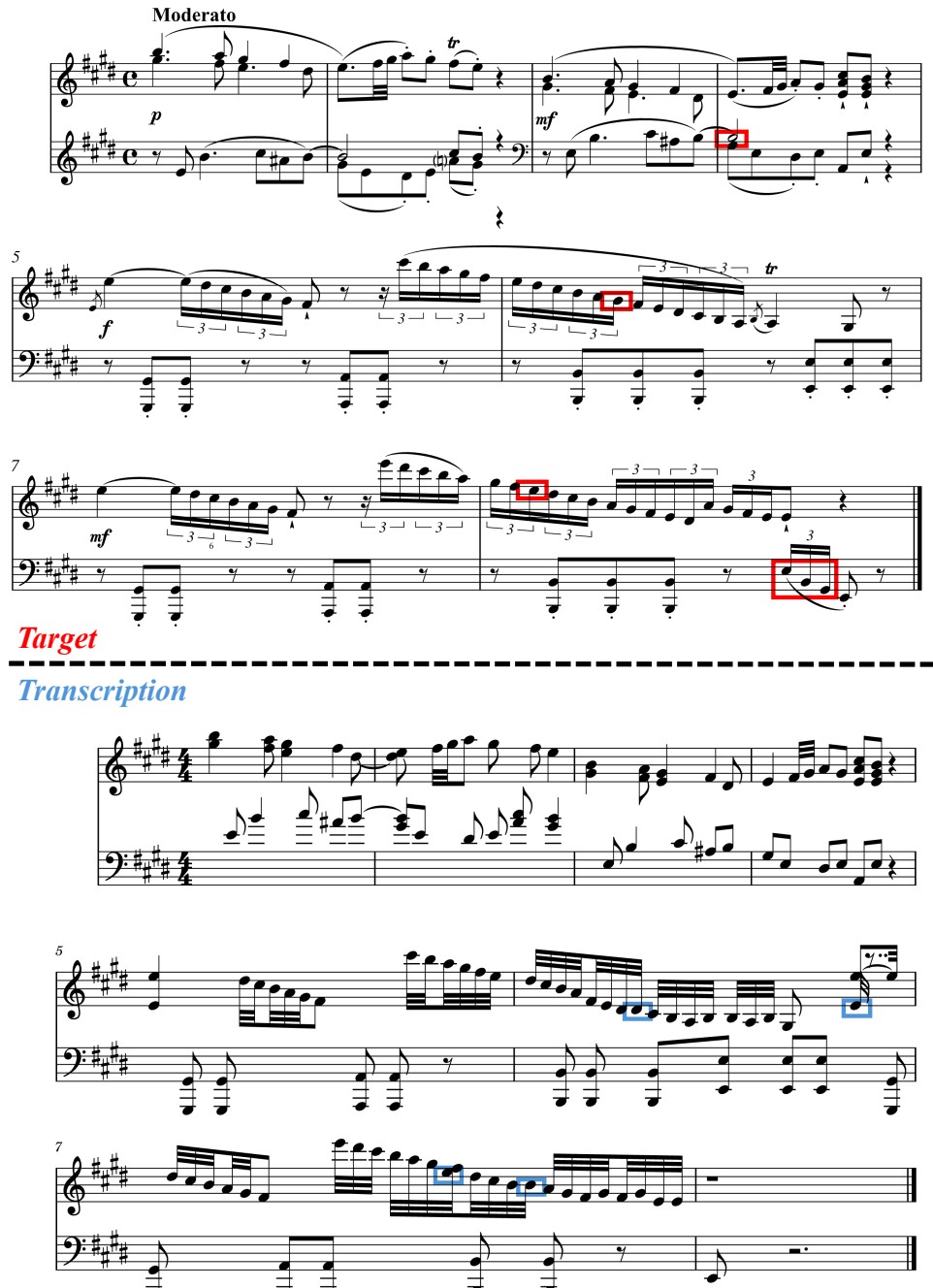

Figure 10: Ground truth score (*upper*) and transcribed score (*lower*) from *Keyboard Sonata in E major, Hob.XVI:31*, by Franz Joseph Haydn (APT sample 2).

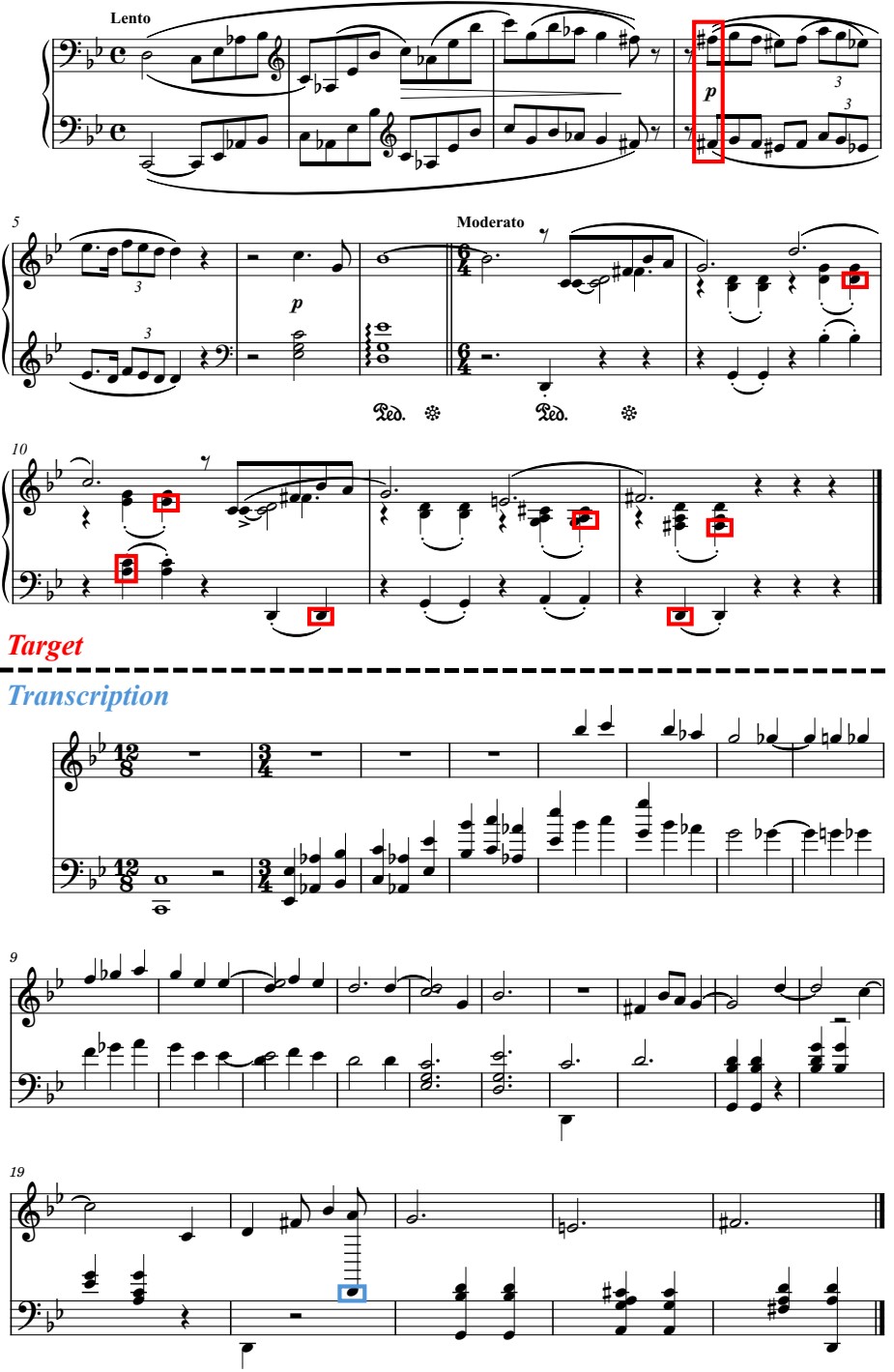

Figure 11: Ground truth score (*upper*) and transcribed score (*lower*) from *Ballade No. 1 in G minor, Op. 23*, by Frédéric Chopin (APT sample 3).

