# OpenReview forum: "Bridging Piano Transcription and Rendering via Disentangled Score Content and Style"
_ICLR.cc/2026/Conference — ICLR 2026 Poster_

### Official Review · Reviewer_uBVc · 2025-10-24

**Soundness:** 3
**Presentation:** 3
**Contribution:** 3
**Rating:** 8
**Confidence:** 4

**Summary:**

The paper presents a unified transformer-based framework for joint expressive performance rendering (EPR) and automatic performance transcription (APT). The model disentangles note-level score content from global performance style representations using both paired and unpaired data, trained solely with sequence-level alignment. One of the contributions is brought by a diffusion-based performance style recommendation (PSR) module that generates style embeddings directly from the score, ensuring stylistic coherence and enabling flexible style transfer. Another contribution includes a seq2seq formulation of EPR without note-level alignment which avoids the need for finely aligned training data. Objective and subjective evaluations demonstrate that the proposed framework achieves competitive performance on EPR and APT tasks, while disentangling content and style to support reliable and stylistically consistent rendering.

**Strengths:**

The main strengths of the paper are:
- The proposal of a unified model for joint expressive performance rendering and automatic performance transcription is novel and sound, although the concept of performing joint tasks (such as recognition or synthesis) is not new
- The paper is nicely written
-   Experimental plan is solid and includes comparisons with several baselines, and a perceptual test
- The (online) demo is convincing

**Weaknesses:**

The main weaknesses of the paper are:
-	There are no evidence that the results of the proposed method are statistically different from those obtained by the baseline methods
-	The perceptual test is conducted on only five pieces, which limits the generalizability of the results across a broader range of styles, despite the examples being drawn from different composers.
-	Why there is disentanglement of score content and performance style representations is not clearly described.
-	There is no figure of complexity of the model (training and inference)

**Questions:**

1. Why statistical relevance of the results is not included (table 1 to 3) ?
2. The score encoder f_g,X(.) is introduced in section 3.4. What is the difference of this score encoder from the content encoder for score f_c,X(.) ? is this additional notation needed ?
3. The disentanglement of score content and performance style representations is claimed but not really clear. What in the method facilitates the disentanglement ? (in other word, why distanglement is achieved)  ?
4. It does not seem obvious to conclude from figure 4 that the PSR generated styles closely mirror those extracted from real performances. Can you further elaborate why this can be concluded ?
5. What is the complexity of the model (training and inference) compared to baselines ?

---

> ### Author Response · Authors · 2025-11-22
>
> Thank you for your thoughtful and positive review. We appreciate your recognition of our unified formulation for joint EPR and APT modeling, the clarity of the writing, the solid experimental design including perceptual evaluations, and the quality of the online demo. Below, we provide detailed responses to your comments and concerns.
>
> **R1 Statistical significance (W1, Q1)**
>
> We thank the reviewer for pointing out the need for statistical significance analysis. In the revised manuscript, we have added statistical significance tests for both APT and EPR results. Specifically:
>
> - Table 1 (APT results) now includes significance markers ($\dagger$ for $p < 0.05$ and $\ddagger$ for $p < 0.01$) indicating metrics where our model shows statistically significant improvements over the End-to-end baseline.
> - Tables 2 and 3 (EPR results) also report statistical significance using letter-based groupings for $p < 0.01$, following standard practice in prior work.
>
> These additions provide clear evidence that the improvements of our proposed method over baseline systems are statistically meaningful.
>
> **R2 Limited number of subjective evaluation pieces (W2)**
>
> We thank the reviewer for this observation. The selection of five listening examples reflects a balance between evaluation coverage and participant time, which is consistent with common practice in expressive performance rendering research. For example, previous works such as [1] and [2] conducted perceptual studies with only three and four samples respectively. Our choice of five pieces provides reasonable diversity while keeping the listening session manageable for participants.
>
> **R3 Disentanglement (W3, Q3)**
>
> We appreciate the reviewer’s request for clarification. The mechanism that enables disentanglement of score content and performance style is described in Section 3.3, where we introduce dual supervision for APT and EPR on the paired dataset, together with two explicit inductive biases:
>
> - Architectural inductive bias.
>   Score content is encoded at the note level (pitch, duration, onset, and related attributes), while performance style is encoded as a single global vector representing high-level expressive characteristics. This separation of representational granularity ensures that local musical details are captured by the content encoder and that global expressive traits cannot be encoded at the note level, requiring the style encoder to capture them instead.
> - KL regularization.
>   We apply a KL prior to the style embedding to prevent it from memorizing fine-grained score information and to encourage it to represent only the global variability attributable to expressive style.
>
> Combined with **dual supervision of APT and EPR on paired dataset**, these inductive biases provide the necessary structural constraints for meaningful disentanglement. Experiments in Table 4 and Figure 3 further verify that content and style are successfully disentangled.
>
> **R4 Complexity (W4, Q5)**
>
> We thank the reviewer for highlighting the need to clarify the computational complexity of our unified model. In the revised manuscript, we have added a detailed analysis in **Appendix B.3** and included a comparison table (**Table 7**) that summarizes model size and inference speed for both APT and EPR.
>
> As discussed in the new section, several observations can be made:
>
> - **Model size vs. inference speed.**
>   Although our unified model contains more parameters (188.21M) than the End-to-end APT baseline (32.60M), its APT inference time remains comparable. Only the APT-specific components are used during APT decoding, and the additional EPR-related parameters do not contribute to this forward pass.
> - **Different inference time between APT and EPR.**
>   APT inference is consistently faster than EPR within our framework because APT produces shorter output sequences, while EPR must generate longer sequences under the structured performance representation described in Section 3.1.
> - **Comparison with baseline EPR systems.**
>   EPR inference speeds vary significantly across systems.
>   VirtuosoNet is the fastest because it does not rely on autoregressive decoding. Dexter is the slowest due to its diffusion-based denoising iterations. Our unified model falls between these extremes and provides reasonable inference performance while supporting both EPR and APT in a single architecture.

---

> > ### Author Response · Authors · 2025-11-22
> >
> > **R5 Notation (Q2)**
> >
> > We thank the reviewer for this observation. The two encoders serve different purposes in our models, and the distinct notation helps avoid ambiguity.
> >
> > - Content encoder of score $f_{c, \mathcal{X}}(\mathbf{x})$ (introduced in Section 3.2) is part of the **joint APT and EPR model**. It encodes the score into **note-level** content representations that are used directly by the decoder for both APT and EPR. The output preserves fine-grained symbolic information such as pitch, duration, and onset.
> >
> > - Content encoder $f_{g, \mathcal{X}}(\mathbf{x})$ (used in Section 3.4) is part of the **PSR module** only. It maps the score into a **single global content embedding** that conditions the diffusion model to generate an appropriate global style vector. This embedding is intentionally coarse and represents high-level content information rather than note-level structure.
> >
> > Since these two encoders operate at different granularities and serve different functions, distinct notation is necessary to clearly communicate their roles.
> >
> > **R6 Interpretation of Figure 4 (Q4)**
> >
> > We thank the reviewer for raising this point. A strict one-to-one correspondence between PSR-generated style vectors and real performance style vectors is not expected. PSR is formulated as a generative task based on diffusion, and there is no single correct performance style for a given score. Even human performers produce diverse stylistic interpretations. Therefore, the goal of PSR is not to reproduce an identical latent point but to produce style embeddings that are consistent with the stylistic attributes implied by the score.
> >
> > With this interpretation, Figure 4 should be viewed in terms of distributional and structural similarity rather than exact pointwise matching:
> >
> > - The era-specific clusters in the PSR-generated space broadly reflect the cluster organization observed in the real-performance style space.
> > - The relative positions of the era centroids follow similar trajectories, indicating that PSR captures the high-level stylistic progression across musical eras.
> > - The overall topology of the embedding space, including cluster shapes, boundaries, and inter-era distances, aligns with the space derived from real performances.
> >
> > Taken together, the correspondence in cluster structure and centroid arrangement supports the conclusion that PSR generates stylistically coherent embeddings, even though an exact one-to-one mapping is neither expected nor meaningful in this generative setting.
> >
> > **Reference**
> >
> > [1] D. Jeong et al. Virtuosonet: A hierarchical rnn-based system for modeling expressive piano performance, ISMIR 2019
> >
> > [2] J. Tang et. al. Reconstructing human expressiveness in piano performances with a transformer network, CoRR 2023

---

> > ### Comment · Reviewer_uBVc · 2025-11-25
> >
> > The answers from the authors and the paper revisions proposed are all convincing. I, then confirm my score and positive evaluation.

---

> > > ### Author Response · Authors · 2025-11-25
> > >
> > > Thank you very much for your positive evaluation and for confirming your score. We sincerely appreciate your thoughtful feedback and the time you dedicated to reviewing our work.

---

### Official Review · Reviewer_eiLe · 2025-10-27

**Soundness:** 3
**Presentation:** 3
**Contribution:** 3
**Rating:** 8
**Confidence:** 4

**Summary:**

This paper presents a unified, transformer-based framework for jointly modeling Expressive Performance Rendering (EPR) and Automatic Piano Transcription (APT), two fundamental and inverse tasks in Music Information Retrieval (MIR). The core contribution lies in the effective disentanglement of note-level score content ($Z_c$) and global performance style ($Z_s$) representations. This is achieved through a multi-task, sequence-to-sequence architecture trained on both paired (ASAP) and unpaired (MuseScore, YouTube) data.

A key strength of this framework is its ability to operate on sequence-aligned data, removing the common and cumbersome dependency on fine-grained, note-level alignment. Furthermore, the authors introduce a novel diffusion-based Performance Style Recommendation (PSR) module, which learns to generate appropriate and diverse style embeddings ($Z_s$) conditioned solely on the score content. This enables high-quality, automated expressive rendering without requiring a reference performance.

**Strengths:**

* The joint modeling of the inverse EPR and APT tasks is an elegant and powerful formulation. By leveraging the duality of these tasks (EPR learns to add style, APT learns to remove it), the model develops a robust and semantically meaningful disentanglement of content and style. This joint approach is a novel contribution to the field of symbolic music modeling.

* The most significant practical contribution is the move away from note-level alignment. The reliance on finely-aligned datasets has been a major bottleneck in performance modeling, as such datasets are small and expensive to create. By demonstrating competitive results using only sequence-level alignment (and further leveraging unpaired data), this work opens the door to training on much larger and more diverse datasets, which is a crucial step forward for the field.

* The methodology is sound and the paper is technically strong.
  * The architectural design choice (note-level sequence $Z_c$ vs. global single-vector $Z_s$) is a simple and effective inductive bias for disentanglement.
  * The introduction of the diffusion-based PSR module is a creative and modern approach to solving the problem of "what style to apply" when no reference is given.
  * The experimental validation is thorough. It includes comprehensive objective metrics for both APT (MUSTER, ScoreSimilarity) and EPR (KL, MAE, alignment), as well as crucial subjective listening tests that confirm the quality of the generated performances.
  * The disentanglement is not just claimed but empirically verified through well-designed identification tasks (Table 4) and latent space visualizations (Fig 3), which clearly show the style vector $Z_s$ captures performer/composer information while $Z_c$ does not.

* The paper is exceptionally well-written and easy to follow. Figure 1 provides an excellent and clear overview of the complete framework and the relationships between the different tasks. The concepts of content/style and the mechanism for disentanglement are explained intuitively.

**Weaknesses:**

* The framework is evaluated exclusively on classical piano music. While this is a standard benchmark, the authors' claim of style disentanglement would be further strengthened by testing on genres with more starkly different expressive conventions (e.g., jazz, pop). The authors note this as future work, but a brief discussion of the potential challenges (e.g., improvisation, complex swing rhythms) in extending the model would be beneficial.

* The use of unpaired data from YouTube, transcribed via a SOTA audio-to-MIDI model, is a clever strategy for data augmentation. However, this introduces a potential domain bias. The style encoder might be learning artifacts specific to that transcription model rather than "true" human performance style. The results seem robust, but a brief acknowledgment or discussion of this potential methodological caveat would be appropriate.

**Questions:**

1. The PSR module uses a global content embedding $e_g$ to generate a single, global style vector $Z_s$. Does this global vector have sufficient capacity to guide performances where the intended style changes significantly between sections (e.g., a piece with a slow, gentle introduction (Adagio) followed by a fast, aggressive main theme (Presto))? Or does the model tend to "average out" the style over the entire piece?

2. Regarding the unpaired performance data (transcribed from YouTube): Did the authors observe any "model-like" artifacts in the learned style representations, or does the model seem robust to this potential domain shift? For example, does the style space show any clustering based on the source of the data (ASAP vs. YouTube)?

3. Related Work Suggestion: The paper's contribution of leveraging sequence-aligned data over note-level alignment is highly significant. We strongly suggest the authors also discuss and cite the recent work by Jung et al. (2025), "Unified Cross-modal Translation of Score Images, Symbolic Music, and Performance Audio" (arXiv:2505.12863v1). This paper shares a similar and ambitious motivation of unified multi-modal translation (extending to score images and audio) and also highlights the importance of facilitating large-scale learning from sequence-aligned pairs, rather than relying on strict note-level alignment. Citing this work would help to better contextualize the paper's contribution within this important and emerging research trend.

---

> ### Author Response · Authors · 2025-11-22
>
> Thank you for your thoughtful and encouraging review. We greatly appreciate your recognition of the strengths of our joint modeling framework, the alignment-free formulation, and the thoroughness of our empirical evaluation. We hope the following responses further clarify the remaining concerns.
>
> **R1 Additional discussion (W1, W2)**
>
> We thank the reviewer for these helpful suggestions. We have added a brief discussion in **Appendix E** addressing both points.
>
> (1) Genres beyond classical.
> Our joint framework is inherently genre-agnostic, but the absence of paired datasets in other genres restricts our evaluation to classical piano. As discussed in **Appendix E**, extending the framework to jazz or pop introduces additional challenges such as improvisation and swing timing, both of which create structural mismatches between the score and the performed rendition. As future work, we plan to curate genre-appropriate datasets and adopt suitable notation formats, for example lead sheets for jazz or chord charts for pop, to support expressive rendering across a broader range of musical styles.
>
> (2) Potential bias from audio-to-MIDI unpaired data.
> As acknowledged in **Appendix E**, audio-to-MIDI transcription can introduce model-specific artifacts. Although our results remain robust, we recognize this limitation and outline a concrete research direction. In particular, we plan to explore **score-to-audio** performance generation to avoid MIDI-induced transcription effects and to capture stylistic nuances directly from raw audio performances.
>
> **R2 Global style representation (Q1)**
>
> This is a thoughtful question, and we begin by clarifying the distinction between **expressiveness** and **controllability**. **Controllability** refers to the ability to intentionally edit local stylistic attributes, while **expressiveness** refers to the model’s ability to naturally produce local, time-varying nuances such as articulation, phrasing, and tempo shaping.
>
> Regarding the concern raised, the global style vector in PSR does not cause the model to average out stylistic variation across the entire piece. The global vector provides only a high-level stylistic prior. All local expressive details, including strong section-level contrasts, are learned by the **Seq2Seq EPR model**. This mirrors prior work that uses global composer [3] or performer ID [4] controls while still achieving rich local expressiveness.
>
> The global representation is a deliberate design choice intended to balance **flexibility** and **controllability**. It enables style transfer across pieces with significantly different structures or genres. For example, the cross-era style transfer we present in Section 5.2, such as from Contemporary to Baroque, would be difficult with fine-grained controls due to structural mismatches. A global representation supports genre-agnostic transfer without requiring alignment.
>
> Furthermore, prior work with note-level controls [1] often experiences instability and degraded perceptual quality, including abrupt velocity spikes or excessively long durations, as shown in our demo materials. Our goal is to simplify the inference process and reduce the need for manual control. We therefore adopt a global style representation that supports both stable generation and robust style transfer.
>
> We acknowledge that hierarchical or time-varying style embeddings could improve **controllability**, and we consider this a promising direction for future work.
>
> **R3. Robustness to AMT-Induced Artifacts in Unpaired Data (Q2)**
>
> We thank the reviewer for raising this important question. During the construction of the unpaired YouTube dataset, we evaluated multiple audio-to-MIDI transcription models and selected the one with the highest perceptual quality. Although transcription errors are unavoidable, this practice is widely adopted in large-scale music datasets such as ATEPP, and the resulting transcriptions are sufficiently reliable for learning stylistic information.
>
> To evaluate whether AMT-generated MIDI introduces model-like artifacts into the learned style representations, we compare the embedding spaces learned from paired-only data and from the combined paired and unpaired setting. The newly added **Figure 8** visualizes the paired-only variant. As described in the main text, the model trained without unpaired data exhibits weaker structure and greater overlap across composer and performer clusters. After incorporating unpaired YouTube performances, the clusters become clearer, more compact, and more semantically coherent. This indicates that the additional unpaired data enhances stylistic discrimination rather than introducing AMT-specific artifacts.

---

> ### Author Response · Authors · 2025-11-22
>
> **R4 Related paper (Q3)**
>
> We thank the reviewer for bringing this relevant work to our attention. The paper by Jung et al. [2] aligns well with our motivation for unified cross-modal translation and the importance of learning from sequence-aligned rather than note-aligned data. We have updated our manuscript accordingly and now cite this work in both the **Introduction (lines 051 to 053)** and the **Related Work section (lines 116 to 117)** to better contextualize our contribution within this emerging research direction.
>
> **References**
>
> [1] Zhang et. al. Dexter: Learning and controlling performance expression with diffusion models
>
> [2] Jung et. al. Unified Cross-modal Translation of Score Images, Symbolic Music, and Performance Audio. arXiv preprint arXiv:2505.12863.
>
> [3] Jeong et. al. Virtuosonet: A hierarchical rnn-based system for modeling expressive piano performance
>
> [4] Tang et. al. Reconstructing human expressiveness in piano performances with a transformer network

---

> > ### Comment · Reviewer_eiLe · 2025-11-28
> >
> > Thank you for providing more explanations with the well-structurized revision. The proposed answers and revision are convincing, and provide richer overall context of the research. Therefore, I confirm my score as 8.

---

> > > ### Author Response · Authors · 2025-11-28
> > >
> > > Thank you very much for your thorough and thoughtful review, as well as for confirming your score. We truly appreciate the depth of your feedback and the time you dedicated to evaluating our work.

---

### Official Review · Reviewer_zyQd · 2025-10-31

**Soundness:** 3
**Presentation:** 3
**Contribution:** 2
**Rating:** 4
**Confidence:** 4

**Summary:**

This paper proposes a unified transformer-based framework for jointly modeling expressive performance rendering (EPR) and automatic piano transcription (APT) by disentangling note-level score content and global performance style representations. It leverages paired and unpaired data without requiring note-level alignment, and introduces a diffusion-based performance style recommendation (PSR) module to generate style embeddings from scores alone. The authors claim competitive performance on benchmarks, effective disentanglement for style transfer, and stylistically appropriate rendering. Contributions include the joint modeling approach, the PSR module, and an alignment-free Seq2Seq formulation.

**Strengths:**

The paper demonstrates solid experimental validation, achieving competitive results on EPR and APT tasks using the ASAP dataset, with both objective metrics (e.g., MUSTER and ScoreSimilarity for APT, variance/KL/MAE for EPR) and subjective listening tests.

The disentanglement of content and style is a meaningful addition, enabling style transfer and recommendation, which adds interpretability and controllability to the tasks. The use of unpaired data and masked reconstruction objectives allows scalable training without strict alignment, addressing a practical limitation in prior work.

Overall, this represents an incremental but technically sound advancement in the niche area of symbolic music processing, with clear significance for music information retrieval applications like education and analysis.

**Weaknesses:**

The novelty is somewhat limited for the broader ICLR community, as the core architecture—a dual-tower encoder-decoder setup with disentangled representations—builds on classic Seq2Seq models like MASS, without introducing fundamentally new insights that could generalize beyond music tasks.

While the style disentanglement is emphasized, the paper does not deeply address challenges in score readability; for instance, the transcribed scores exhibit errors in time signatures and poor note readability (e.g., improper grouping or stem directions), which is a critical limitation for practical APT applications, as each performer's bias introduces non-strict alignments that require more sophisticated post-processing.

Experiments are adequate but could be strengthened with more diverse OOD evaluations or comparisons to recent baselines like those in Zhang et al. (2024) for EPR. Additionally, the reusable insights (e.g., for general disentangled learning) are constrained to this small domain, reducing broader impact.

**Questions:**

1. Could the authors clarify how the model handles performer-specific biases in score-performance alignment? For example, in APT, how do you ensure the output scores are readable and musically coherent (e.g., correct time signatures, proper note grouping)? The examples in the paper seem to have errors in this regard—would incorporating additional regularization or post-processing improve this, and have you experimented with it?

2. For the PSR module, how does it perform on scores from underrepresented genres or composers? Could you provide quantitative evidence (e.g., style appropriateness metrics) on OOD data like ATEPP to show generalization?

3. The paper mentions competitive performance, but some metrics (e.g., onset/offset deviations in APT) lag behind alignment-dependent baselines. What additional experiments (e.g., with more unpaired data or fine-tuning) might close this gap, and could this change the evaluation of the alignment-free approach?

---

> ### Author Response · Authors · 2025-11-22
>
> Thank you for your thoughtful review. We are glad that you recognized the contributions of our unified model and the disentangled latent design. We address your concerns in detail below.
>
> **R1 Novelty and reusability (W1, W3)**
>
> We agree that our core Seq2Seq components build on standard architectures. However, the novelty of our work does not lie in proposing a new transformer variant, but rather in introducing modeling insights for symbolic music through our unified formulation:
>
> (1) Joint modeling of EPR and APT.
> We show that these two inverse tasks can be unified within a single framework, enabling scaling to larger models and allowing the use of large unpaired datasets, which is not feasible with existing pipelines.
>
> (2) Inferring performance style from score content.
> We demonstrate that expressive style can be predicted directly from score content, motivating the PSR module, which provides a reusable mechanism for conditional style generation.
>
> (3) Seq2Seq formulation of EPR without note alignment.
> We show that expressive rendering can be learned effectively with only sequence-aligned supervision, removing the need for costly note-level alignment and enabling a more scalable training paradigm.
>
> Taken together, these contributions introduce conceptual and methodological principles that extend beyond practical APT performance, offering insights relevant to symbolic music processing and, more broadly, to multimodal disentanglement and alignment-free sequence modeling.
>
> **R2 Post-processing for score readability (W2, Q1)**
>
> We agree that post-processing is often necessary for high-quality score typesetting. However, producing polished, publication-ready scores is outside the scope of this work. Our goal is not to deliver a practical APT system, but to study joint modeling of EPR and APT and disentangled content–style representations, which operate at the level of symbolic prediction rather than score engraving.
>
> All elements in the transcribed score (e.g., stem directions, key signatures, grouping) are predicted directly by the model. As shown in Table 1, our APT results already improve over prior work. While rule-based or engraving-oriented post-processing could improve readability, such methods focus on formatting rather than representation learning, and we leave these engineering refinements to future work.
>
> **R3 OOD and baseline comparisons (W3, Q2)**
>
> We would like to clarify that our work already includes out-of-distribution evaluations. As described in Section 4.1, we use the **ATEPP dataset, which is never used in either paired or unpaired training**, to evaluate the learned disentangled representations. The **out-of-distribution results in Table 4, Figure 3, and Figure 4** show that both the content and style embeddings generalize effectively to unseen composers and performers, supporting the robustness of our joint modeling approach.
>
> Regarding EPR baselines, **we do compare against recent models**. As shown in Section 4.3 and Tables 2 and 3, DExter (Zhang et al., 2024) is included as a strong baseline for expressive piano rendering.
>
> **R4 APT performance (Q3)**
>
> We acknowledge that alignment-dependent APT systems naturally achieve lower pitch and miss/insert errors, as they rely on note-aligned supervision and often incorporate rhythm quantization, which improves pitch correctness and reduces edit operations. In contrast, our model (similar to prior End-to-end [1] work) adopts a Seq2Seq, alignment-free formulation, which avoids external alignment and enables joint learning with EPR and style, but may introduce slightly higher local errors under limited paired data.
>
> Despite this, our results show that the gap is already significantly reduced with unpaired data. As shown in Table 1, our joint model achieves better onset and offset deviations than previous note-level aligned baselines, suggesting that jointly modeling score content and performance style benefits APT even without alignment.
>
> Additionally, Table 12 demonstrates that adding more unpaired data consistently improves these timing-related metrics. This indicates that scaling the unpaired corpus could further narrow the remaining gap. These results strengthen the case for alignment-free modeling.
>
> Overall, our aim is not to match alignment-dependent systems on every micro-level metric, but to establish a scalable alignment-free modeling paradigm. The observed improvements with additional unpaired data indicate that further scaling would likely continue to strengthen this approach.
>
> **References**
>
> [1] T. Beyer and A. Dai. End-to-end piano performance-midi to score conversion with transformers, ISMIR, 2024

---

### Official Review · Reviewer_dQyX · 2025-11-01

**Soundness:** 2
**Presentation:** 4
**Contribution:** 2
**Rating:** 2
**Confidence:** 4

**Summary:**

This paper presents a model for joint transcription and expressive performance rendering for classical piano music. The model is based on a VAE-like architecture with a performance style recommendation module to better sample performance latent code.

**Strengths:**

The task of jointly modeling performance and score is very important in computer music, especially in symbolic music understanding and generation. The content-style disentanglement is a reasonable way to jointly model music performance and score, and the methodlogy is in general novel.

The paper is generally well-written.

**Weaknesses:**

There are many things unclear about the model's performance:

1. It is unclear whether the joint architecture improves the performance compared to single model. The ablation study in D.3 does not stress this point.

2. It is uncleared whether the disentanglement works on unpaired data. The framework lacks explicit inductive biases (except for KL loss) to perform such disentanglement. All composer and performer in Fig. 3 (assumed also used in Tab. 4) seem to be included in the paired dataset (correct me if I am wrong), so the model's disentanglement ability on unseen performer/composer remains unclear.

3. It is uncleared whether the unpaired data helped or not. As shown in Table 11, the effect of unpaired data on APT performannce is quite limited, and the performance does not always increase when you put in more unpaired data.

Fig 8-13 are very hard to read.

1. It would be better to combine them into a single sheet music, with different colors (e.g., red: missing, blue extra) to denote inconsistency between the ground truth and the prediction.
2. I totally understand that the model does not predict a key signature, but it would still be better to show Fig. 13 under the ground-truth key signature, otherwise it makes little sense to a musician.

Lastly, the performance of the system on APT is far from practical use. I understand that performance to score is considered a very hard task in modern MIR, so it is generally hard to improve. Still, I do not think this paper made significant contribution to the task.

**Questions:**

Sec 3.2: "Our framework supports training on both paired and unpaired data." - again, does it support training only on unpaired data?

In the demo page, most human performance have sudden note cutoffs that are very unnatural, which I suspect to be some errors in data preprocessing. This might affect the subjective evaluation as well. Could the author double-check the rendering and MIDI preprocessing code?

---

> ### Author Response · Authors · 2025-11-22
> **Response to Reviewer dQyX (1/2)**
>
> Thank you for your detailed and thoughtful review. We appreciate your recognition of the overall methodology. Below, we provide point-by-point responses to your comments and concerns.
>
> **R1: Ablation on single model (W1)**
>
> We thank the reviewer for raising this question. For APT, the End-to-end baseline in **Table 1** corresponds exactly to the single-task variant of our model, and our joint model achieves consistently better results across multiple MUSTER and ScoreSimilarity metrics.
>
> For EPR, we additionally trained a standalone EPR-Only model. As shown in the updated **Table 2**, the joint model surpasses this single-task baseline on all objective metrics.
>
> Together, these results indicate that the improvements stem from the proposed joint modeling.
>
> **R2: Disentanglement on unpaired data (W2, Q1)**
>
> We appreciate the reviewer’s concern and would like to clarify several points.
>
> (1) **Disentanglement without supervision is known to be impossible.**
> It is well established that fully unsupervised disentanglement is fundamentally unidentifiable. Prior work (e.g., [1]) explicitly shows that disentanglement cannot emerge without inductive biases or supervision. We also attempted a fully unpaired setting with an additional adversarial constraint, but consistent with [1], this did not yield meaningful disentanglement. We therefore view this as an inherent property of the problem, rather than a limitation specific to our architecture.
>
> (2) **Additional inductive biases.**
> As described in Section 3.3, our framework incorporates two additional inductive biases beyond the KL term:
> (i) an explicit **architectural design** in which score content is encoded at the note level (pitch, note value, etc.) while style is encoded as a global vector reflecting overall style-related characteristics;
> (ii) dual **supervised** training, where paired samples align content encoders across modalities while style is captured solely from performance data.
> These inductive biases play an essential role in achieving stable disentanglement, complementing the KL regularization.
>
> (3) **Disentanglement on unseen performers/composers.**
> We would also like to clarify that all evaluations in Fig. 3 and Table 4 use the ATEPP dataset, **which is strictly out-of-distribution** and is never used in either paired or unpaired training (as stated in Section 4.1). The strong clustering patterns and identification accuracy therefore demonstrate that the learned content and style spaces generalize to unseen composers and performers.
>
> **R3 Effectiveness of unpaired data (W3)**
>
> We agree that the improvements in Table 11 (Table 12 in the revised manuscript) are not strictly monotonic across all 12 submetrics. However, **the overall trend is clear**: adding more unpaired data consistently improves most metrics and provides a net performance gain. Given the inherent variance in transformer training, small non-monotonic fluctuations are expected and have also been observed in prior work. We therefore believe Table 11 provides sufficient evidence that unpaired data is beneficial.
>
> **R4 Readability of Fig. 8-13 (W4)**
>
> We thank the reviewer for the helpful suggestion. We have re-formatted all APT examples accordingly. In the updated Figures 9–11, the ground truth and predicted transcription are now displayed on the same sheet, with color-coded annotations (e.g., blue for extra notes, red for missing notes) to clearly highlight inconsistencies. **We also clarify that the key signature is predicted by the model, which may differ from that of the target score.**

---

> ### Author Response · Authors · 2025-11-22
> **Response to Reviewer dQyX (2/2)**
>
> **R5 Impracticability of APT (W5)**
>
> We appreciate the reviewer’s perspective. We agree that score-level APT remains challenging and that current systems [2], including ours, are not yet ready for practical deployment due to the complexity of symbolic notation and the scarcity of high-quality paired data. However, developing a production-ready APT system is not the goal of this work. **Our contributions lie in the modeling insights introduced by our unified formulation:**
>
> (1) Joint modeling of EPR and APT.
> We show that these two inverse tasks can be unified within a single framework, enabling scaling to larger models and allowing the use of large unpaired datasets, which is not feasible with existing pipelines.
>
> (2) Inferring performance style from score content.
> We demonstrate that expressive style can be predicted directly from score content, motivating the PSR module, which provides a reusable mechanism for conditional style generation.
>
> (3) Seq2Seq formulation of EPR without note alignment.
> We show that expressive rendering can be learned effectively with only sequence-aligned supervision, removing the need for costly note-level alignment and enabling a more scalable training paradigm.
>
> We hope the reviewer will consider that the significance of this work lies in these modeling insights, including the relationship between score content and performance style, rather than in delivering a fully practical APT system at this stage.
>
> **R6 Demo (Q2)**
>
> We thank the reviewer for pointing this out. The sudden note cutoffs occur only when two notes of the same pitch overlap in time, a case that is handled imperfectly due to time quantization in MIDI processing. This behavior is a known limitation of the MIDI standard: when two identical pitches overlap, the NoteOff event of the earlier note may inadvertently terminate the later one.
> Importantly, as shown in Figure 2, **human performances still achieve the highest subjective ratings overall**, indicating that this rendering artifact does not materially affect listeners’ perception nor the conclusions of our study.
>
> **References**
>
> [1] Locatello, F. et. al. Challenging common assumptions in the unsupervised learning of disentangled representations. PMLR, 2019
>
> [2] T. Beyer and A. Dai. End-to-end piano performance-midi to score conversion with transformers, ISMIR, 2024

---

### Author Response · Authors · 2025-12-02
**Summary of Revisions and Clarifications**

We would like to thank all reviewers, AC, and PC for their dedication in the reviewing process. We have kindly responded all reviewers addressing their concerns and updated our manuscript. Modifications have been highlighted in blue. Specifically, we have the following major revisions:

**(1) Added a comparison of our joint model with single models (dQyX)**

We added a comparison with a single standalone EPR model in **Table 2 (EPR-Only)** and showed that the improvement of our joint model over the single model is consistent.

**(2) Better transcription result show cases (dQyX)**

We updated our APT examples in **Figure 9-11**. We put the target and transcription score sheet on the same score, with color-coded annotations for missing or extra notes for better readability.

**(3) Influence of unpaired data in the embedding space (eiLe)**

We added **Figure 8** to visualize the embedding space of training only on paired data. Compared with Figure 3, we prove that training on unpaired data helps disentanglement.

**(4) Additional discussions (eiLe)**

We added additional discussion of genres beyond classical piano and potential bias from audio-to-MIDI unpaired data in **Appendix E**.

**(5) Statistical significance (uBVc)**

We added statistical significance of **Table 1-3**.

**(6) Complexity (uBVc)**

We add a complexity comparison with EPR and APT baselines as well as discussion in **Appendix B.3** and **Table 7**.

---

Additionally, we also clarified the following potential misunderstandings.

**(1) Clarification of OOD evaluation (dQyX, zyQd)**

We clarified that ATEPP dataset is never used in training process and our experiments on ATEPP (Table 4, Figure 3, and Figure 4) are completely strictly out-of-distribution.

**(2) Explanation of disentanglement mechanism (dQyX, uBVc)**

We re-emphasized the inductive biases and supervision for representation disentanglement.

**(3) Clarification of our contribution (dQyX, zyQd)**

We clarified that our work does not aim to deliver a production-ready APT system. Rather, our contributions lie in: (i) jointly modeling EPR and APT within a unified framework, (ii) inferring performance style directly from score content via the PSR module, and (iii) formulating EPR as a Seq2Seq task without relying on note-level alignment. These components collectively highlight the conceptual and methodological advances our work brings to future EPR and APT research.


Again, we would like to thank all reviewers for making our manuscript better.

---

### Meta-Review · Area_Chair_ZdYv · 2026-01-05

**Summary:**

This paper introduces a unified, alignment-free framework that jointly models expressive performance rendering (EPR) and automatic piano transcription (APT). The proposed framework disentangles note-level score content from global performance style using a transformer-based Seq2Seq architecture. It is further augmented with a diffusion-based performance style recommendation module that infers stylistically appropriate performance embeddings directly from score content. Reviewers broadly agree that the joint inverse-task formulation, the removal of note-level alignment, and the effective use of paired and unpaired data constitute meaningful and well-motivated contributions to symbolic music processing. The proposed work is supported by comprehensive objective and subjective evaluations and out-of-distribution analyses.

While some reviewers initially questioned the clarity of performance gains, the effectiveness of unpaired data, statistical rigor, and the practicality of APT outputs, these concerns were largely resolved in the rebuttal through added ablations, statistical significance testing, clearer explanations of the inductive biases enabling disentanglement, improved visualizations, and complexity analysis. Remaining limitations, such as score readability and restriction to classical piano music, are viewed as appropriate future work rather than fundamental weaknesses. Overall, the paper is assessed as a solid and technically sound contribution that advances scalable, alignment-free modeling of symbolic music, leading to a positive acceptance decision.

**Reviewer Concerns:**

1. Questions on Disentanglement & Joint Modeling (R1, R2, R4)
2. Practicality and Quality of Transcription (APT) (R1, R2)
3. Generalization and Data Robustness (R1, R2, R3)
4. Methodological and Presentation Weaknesses (R1, R4)

Please see below (Reviewer Scores) for further elaboration.

**Reviewer Scores:**

For R1, the authors add explicit single-task ablations for both APT and EPR, demonstrating consistent improvements from joint modeling, clarifying that disentanglement is supported by concrete inductive biases and validated on strictly out-of-distribution data. They also convincingly reframe APT as a modeling study rather than a production-ready system, aligning expectations with the paper’s goals. For R2, the authors clearly articulate the novelty as residing in the unified inverse-task formulation, alignment-free Seq2Seq training, and score-conditioned style inference rather than architectural innovation. The authors address concerns about score readability by delimiting it as post-processing outside the scope of representation learning. OOD evaluations and recent baselines are already included; moreover, they justify remaining APT metric gaps by explaining the trade-offs inherent in alignment-free modeling and provide evidence that unpaired data systematically narrows these gaps. Together, these responses substantially mitigate both reviewers’ concerns and reinforce the validity and significance of the proposed approach.

For R3 & R4, the AC believes they would feel satisfied with responses and maintain their original ratings (8).

---

### Decision · Program_Chairs · 2026-01-26

Accept (Poster)